# Closed-loop control of gamma oscillations in the amygdala demonstrates their role in spatial memory consolidation

Vasiliki Kanta [1,2], Denis Pare[2] & Drew B. Headley[2]

Gamma is a ubiquitous brain rhythm hypothesized to support cognitive, perceptual, and mnemonic functions by coordinating neuronal interactions. While much correlational evidence supports this hypothesis, direct experimental tests have been lacking. Since gamma occurs as brief bursts of varying frequencies and durations, most existing approaches to manipulate gamma are either too slow, delivered irrespective of the rhythm's presence, not spectrally specific, or unsuitable for bidirectional modulation. Here, we overcome these limitations with an approach that accurately detects and modulates endogenous gamma oscillations, using closed-loop signal processing and optogenetic stimulation. We first show that the rat basolateral amygdala (BLA) exhibits prominent gamma oscillations during the consolidation of contextual memories. We then boost or diminish gamma during consolidation, in turn enhancing or impairing subsequent memory strength. Overall, our study establishes the role of gamma oscillations in memory consolidation and introduces a versatile method for studying fast network rhythms in vivo.

[1] Behavioral and Neural Sciences Graduate Program, Rutgers University–Newark, 197 University Ave, Newark, NJ 07102, USA. [2] Center for Molecular and Behavioral Neuroscience, Rutgers University–Newark, 197 University Ave, Newark, NJ 07102, USA. Correspondence and requests for materials should be addressed to D.P. (email: pare@newark.rutgers.edu) or to D.B.H. (email: dbh60@newark.rutgers.edu)

Neuronal networks produce oscillatory states that support local processing and inter-regional communication[1–3]. A particularly widespread rhythm is gamma (30–110 Hz), which is posited to underlie numerous cognitive processes such as attention, working memory, and learning[4]. Although supported by substantial correlative evidence[5–9], hypotheses on gamma function have not been tested directly in vivo due to technical limitations. While available methods can impose gamma rhythms with either tonic or patterned optogenetic stimulation[6,10–12], these manipulations have significant off-target effects, such as altering overall neuronal excitability. Moreover, they do not offer a way to suppress gamma, limiting our ability to assess the rhythm's potential causal relationship with cognitive phenomena. Lastly, since gamma oscillations occur as short intermittent bursts, manipulations that tonically boost the rhythm force the underlying network into an unnatural state.

One of the brain areas that displays prominent gamma oscillations is the BLA, where the rhythm is enhanced during emotional arousal[7,9,13]. The BLA has long been implicated in the emotional modulation of memory consolidation, with manipulations of post-training BLA activity altering the consolidation of aversive[5,6,14,15] and appetitive memories[16–18]. Furthermore, studies have pointed to the role of BLA oscillations in fear expression[9,19]. The presence of gamma oscillations in a structure involved in memory consolidation raises the question whether gamma supports the consolidation process.

To examine the role of gamma in memory consolidation, we introduce a method for precisely modulating this rhythm in real time. We reasoned that since particular phases of the gamma cycle are associated with different levels of neuronal excitability[20], delivery of brief excitatory optogenetic stimuli in-of-phase or out-of-phase with the preferred firing phase of BLA neurons should, respectively, enhance or dampen gamma. Furthermore, since gamma occurs as intermittent bursts[13,21], we should restrict these bouts of stimulation to periods when gamma power is elevated. Thus, we used a programmable signal processor to track the amplitude and phase of ongoing gamma cycles with millisecond accuracy and deliver precisely phase-locked optogenetic stimuli to enhance or suppress them. This allowed us, for the first time, to directly link gamma activity to a cognitive process, namely memory consolidation.

We first establish that BLA gamma oscillations are enhanced in rats shortly after training on Inhibitory Avoidance (IA), a contextual fear task that has been used extensively to study the role of the BLA in enhancing the consolidation of emotional memories[6,14]. We then show that gamma is also increased immediately following training on the hole-board task (HB), a dry, appetitive version of the Morris water maze[22]. Since subjects could be run on this task multiple times, we used it to test the importance of gamma by bidirectionally modulating endogenously occurring gamma oscillations in the BLA after learning. Doing so, we were able to boost or depress memory strength.

## Results

**Higher BLA gamma during consolidation of emotional memories.** During IA training, rats were placed in a brightly lit compartment. Upon entering an adjacent dark section, a shock was delivered. Two days later, rats were again placed in the lit side and latency to enter the dark compartment was measured (IA retention; Fig. 1a). Longer latencies indicated avoidance of the dark compartment and better retention of the shock context. Based on the latency change from training to retention, rats were divided into Good and Poor Learners (Fig. 1b, c). Good Learners displayed a post-training increase in BLA mid-gamma power (40–70 Hz; Supplementary Fig. 1a, b) that lasted for ≥20 min

immediately following training (Fig. 1d) and was significantly more pronounced than in Poor Learners (Fig. 1e). Moreover, peak pre-to-post-training changes in mid-gamma correlated with performance (Fig. 1f) and were not associated with differences in behavioral states (Supplementary Fig. 1c) or motor activity/locomotion (Supplementary Fig. 1d, e). Since gamma oscillations tend to occur as brief bursts, we then examined the effects of IA training at the level of individual mid-gamma bursts (Supplementary Fig. 2). This analysis revealed that changes in the incidence of gamma bursts correlate with IA performance (Supplementary Fig. 2c), whereas their duration and amplitude changes do not (Supplementary Fig. 2a, b).

Next, we tested whether similar changes occur during the consolidation of appetitive spatial memories using the HB task. In this task, each day rats used distal cues to memorize the location of a food reward sitting atop one of five open sand wells, presented on a single training trial (Fig. 1g, h, top). One hour later, three retention trials were conducted separated by 5 min, with the food reward either buried in the original sand well or absent (Probe trial; Fig. 1h, bottom). Latency to visit the correct location decreased significantly from training to retention trials (Fig. 1i), indicating that rats rapidly formed a spatial memory. With longer training-retention intervals, performance degraded (Fig. 1j). As in IA, BLA mid-gamma power increased post-training (Fig. 1k), but the change was weaker (Fig. 1l) and did not correlate with performance (Fig. 1m). Changes in individual burst properties also did not correlate with HB performance (Supplementary Fig. 2d–f). However, the power increases seen in both tasks occured in a similar frequency band with the same half-width (Supplementary Fig. 1f, g).

**BLA gamma is locally generated and coherent in the nucleus.** In order for our closed-loop gamma manipulations to be effective, the gamma oscillations recorded in the BLA have to entrain unit activity, be locally generated, and have a uniform phase throughout the nucleus. While it was repeatedly reported that gamma oscillations entrain unit activity[7,9,13,17], earlier studies did not directly test whether the gamma recorded in the BLA is in fact locally generated. To address this question, we performed current source density (CSD) analysis in two separate animals implanted with multishank silicon probes in the BLA and adjacent regions such as the central amygdala (CeA) and striatum (STR). Consistent with prior findings[13], these recordings verified that multiunit activity (MUA) at all BLA sites was entrained to mid-gamma (Fig. 2a). In particular, MUA was strongest during the trough of gamma, which corresponds to the depolarizing phase of the oscillation, when unit activity should be at its strongest for a locally generated gamma rhythm. Furthermore, mid-gamma bursts were more frequent (Fig. 2b) and stronger (Fig. 2c) in the BLA than at adjacent STR and CeA sites. In addition, gamma bursts were highly synchronized and phase-coherent along the dorsoventral and mediolateral axis of the BLA (Fig. 2d,e; Supplementary Fig. 3), consistent with a prior unit recording study[13], which showed that gamma also has a uniform phase rostrocaudally. This coherence was reduced outside of the BLA, at adjacent cortical and CeA/STR sites (Fig. 2d, e). Finally, we exploited the 8 × 8 array of recording sites on our silicon probe to calculate the CSD power spectrum, which reflects local sinks and sources. Gamma power in the CSD spectrum was much larger in the BLA than in adjacent CeA and STR (Fig. 2f, g).

**A new closed-loop method for real-time gamma modulation.** The increase in BLA mid-gamma after training on both tasks suggests this oscillation might support memory consolidation. To test this hypothesis, we manipulated mid-gamma after training

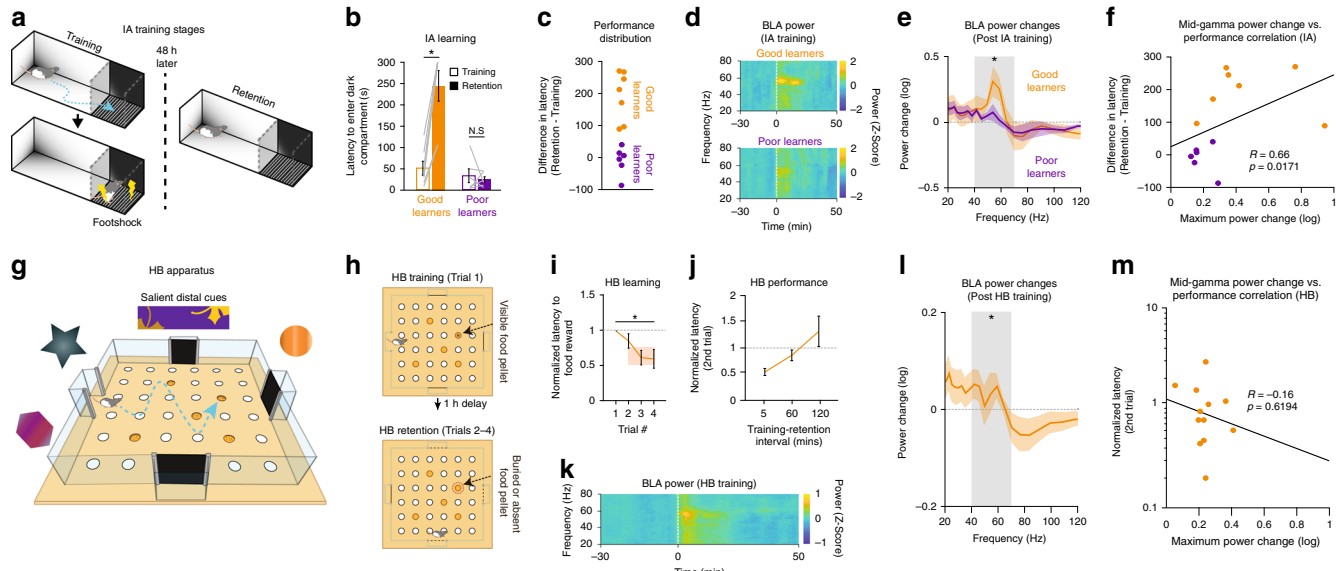

**Fig. 1** BLA gamma power is elevated during consolidation of emotional memories. **a** IA training stages. **b** Latency to enter dark compartment during training (empty bars) and retention (filled bars) for Good (orange, $n = 7$ rats) and Poor Learners (purple; $n = 6$ rats; Two-way ANOVA, main effect of group $F_{(1,25)} = 12.94$, $p = 0.0016$, main effect of session $F_{(1,25)} = 7.4$, $p = 0.0125$, Group × Session interaction $F_{(1,25)} = 11.48$, $p = 0.0026$). Gray lines: individual rats. **c** Performance distribution of IA subjects. **d** BLA power spectrograms before and after IA training, for Good (top) and Poor (bottom) Learners. **e** Average power change (30 min Post-Training minus Pre-Training) for Good (orange) and Poor Learners (purple; Mann Whitney $U(12) = 66$, $p = 0.014$). **f** Correlation between mid-gamma power change and IA performance ($n = 13$ rats, Spearman's rank-order correlation $R = 0.66$, $p = 0.0171$). **g** HB apparatus. **h** HB training stages. **i** Normalized latency to food reward across trials ($n = 13$ rats; Kruskal Wallis ANOVA, $\chi^2(4) = 82.05$, $p = 6.4 \times 10^{-17}$). Shaded orange region reflects Probe trial latency. **j** Normalized 2nd trial latency for different Training-Retention intervals. **k** BLA spectrogram before and after HB training. **l** Average power change in HB task ($n = 49$ sessions; 13 rats; Sign test $t(48) = 12$, $p = 4.88 \times 10^{-4}$). **m** Correlation between maximum mid-gamma power change and HB performance ($n = 13$ rats; Spearman's rank-order correlation $R = -0.16$, $p = 0.619$)

and assessed subsequent memory. We used the HB task for this experiment because it allows multiple tests in the same subjects, therefore enabling within-subject comparisons.

We had to devise a new method to manipulate gamma oscillations in vivo. Previously, gamma was enhanced or imposed by delivering tonic[11], ramped[23], pulse train[6], or stimulus-locked[10] optogenetic stimulation, each of which also causes non-specific increases in network excitability. Another drawback was that there was no way to decrease gamma, apart from silencing all network activity. Closed-loop approaches provide a way around these limitations. Gamma occurs in brief bursts, during which neurons alternate between periods of increased and decreased excitability[3,13], coinciding with the trough and peak gamma phases, respectively. By stimulating neurons during particular gamma phases in a closed-loop manner, it was hypothesized that the rhythm could be enhanced or suppressed[24], a notion supported by in vitro investigations[25]. However, due to processing limitations, previous in vivo closed-loop approaches for bidirectionally modulating endogenous rhythms have only been applied to slow oscillations[26,27]. Only one study performed closed-loop phase-locked tracking of fast rhythms, and it was done in vitro[25].

To overcome the limitations that plagued earlier studies, we used programmable signal processors termed field programmable gate arrays (FPGAs). In contrast with conventional computer operating systems, which perform various background tasks, FPGAs only instantiate dedicated circuits for a desired computation, allowing them to perform signal processing in real time. Taking advantage of this property, we implemented a custom-made algorithm (see Methods section) that tracks the amplitude and phase of ongoing mid-gamma cycles in the LFP with millisecond accuracy. Since particular phases of gamma in the LFP are associated with high and low network excitability[13],

delivering brief excitatory optogenetic stimuli in-phase with high network excitability should enhance the rhythm, while out-of-phase should diminish it. In this method, BLA LFPs are bandpass filtered at mid-gamma (40–70 Hz). When mid-gamma amplitude crosses a threshold and the algorithm detects a selected phase, a 2-ms light stimulus is delivered (Fig. 3a). For an artificial sinusoidal signal (Fig. 3b), our system accurately detected the frequency (Fig. 3c) and phase (Fig. 3d, e). Similar results on spectral specificity were obtained with in vivo LFPs (Fig. 3f, g). As expected, triggering pulses were preceded by mid-gamma activity, and longer bursts contained more triggers (Fig. 4a). On the other hand, there was no change in mid-gamma activity surrounding randomly distributed triggers, which were delivered irrespective of mid-gamma power and phase (Fig. 4b). Furthermore, triggering probability was positively modulated by both the duration and the strength of mid-gamma events (Fig. 4d; Supplementary Fig. 4), something that was not true for other frequency bands (Fig. 4c, e) or random triggering (Fig. 4f–h).

**In vitro validation of our optogenetic approach.** To maximize the temporal precision of our method, we infected BLA neurons unilaterally with Chronos (AAV5.hSyn.Chronos-GFP; Fig. 5a; Supplementary Fig. 5), a fast channelrhodopsin variant[28]. Surprisingly, given that Synapsin is a ubiquitous promoter, in vitro whole-cell recordings (Fig. 5b–i) revealed that Chronos activation by blue light exerted predominantly inhibitory effects in principal neurons (PNs) of the BLA. That is, light stimuli excited most fast-spiking interneurons (FSs; Fig. 5d, e, i; see Supplementary Fig. 6 *for response types*), which elicited picrotoxin-sensitive inhibitory synaptic potentials in PNs (Fig. 5d, f, g). This tropism for FSs was previously reported in somatosensory cortex[29] and in our case was an advantageous result, since FSs are particularly important

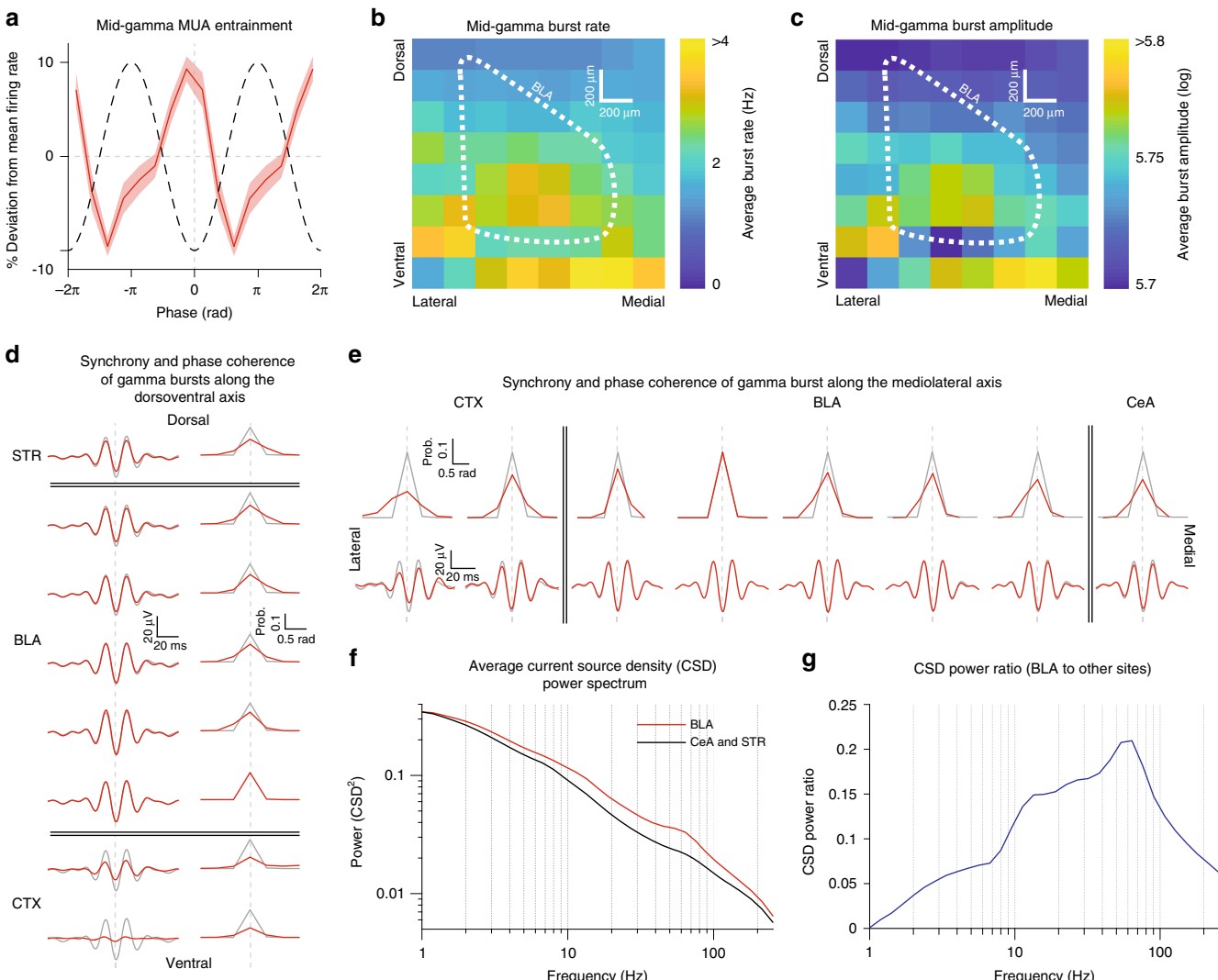

**Fig. 2** Mid-gamma oscillations are locally generated in the BLA and are coherent across its dorsoventral and mediolateral axes. **a** Average multiunit activity (MUA) entrainment across all BLA sites in the 8 × 8 silicon probe. Red: % deviation from average firing rate. Black: Schematic of gamma cycle. **b** Heatmap of mid-gamma burst incidence across all silicon probe sites. X-axis: lateral to medial, Y-axis: ventral to dorsal. Color bar: Average burst rate in Hz. White dashed outline: BLA. **c** Same as in 2b but for mid-gamma burst amplitude. **d** Left: Average raw, unfiltered LFP traces from different dorsoventral recording sites, triggered on the occurrence of mid-gamma bursts at a reference BLA site. Right: Average mid-gamma phase probability at different dorsoventral recording sites, triggered on the occurrence of mid-gamma bursts at a reference BLA site. Red traces correspond to the LFP/phase probability at the corresponding recording site. Right traces show the LFP/phase probability at the reference site, which is the most ventral BLA site. Total number of bursts averaged = 51964. **e** Same as 2d but for mediolateral recording sites. Reference site is the second most lateral BLA site. Total number of bursts averaged = 51964. **f** Average Current Source Density (CSD) power spectrum derived from multishank silicon probe recordings in the BLA (red line, n = 53 sites), as well as central amygdala, and striatum (CeA and STR, black line, n = 46 sites). **g** Ratio of CSD power at sites within versus outside the BLA. The ratio was calculated as follows: (Power in BLA sites–Power in outside sites)/(Power in BLA sites + Power in outside sites)

for gamma generation[8]. In vivo recordings also indicated that the inhibitory interneuron network was recruited by our optogenetic stimulation (Fig. 5j, k). Light pulses caused a transient increase of BLA multiunit activity (MUA) followed by a suppression (Fig. 5l). This coincided with an extracellular positive potential that lasted ~20 ms. This potential likely reflected neural sources since it was abolished by isoflurane overdose (Fig. 5m).

**Real-time modulation bidirectionally affects BLA gamma.** Next, we modulated BLA mid-gamma unilaterally for 1 h after training in the HB task (between Trial 1 and 2). We chose this time window because consolidation of recently formed memories is particularly vulnerable to manipulations of neuronal activity performed in the first few hours after a new experience[30–32].

Moreover, our recordings indicated that this period also exhibited elevations in mid-gamma power (Fig. 1k).

Virus-injected rats implanted with optrodes in the BLA (Supplementary Fig. 5) were first placed in a familiar box ("home box") and recorded for 1 h without stimulation (Light OFF; Fig. 6a). Animals were then trained (Trial 1) and immediately placed back in the "home box" for 1 h (Light ON; Fig. 6a), with one of four different treatments: light pulses were either delivered (1) at the trough of mid-gamma cycles (Trough treatment), (2) at the peak of mid-gamma cycles (Peak treatment), (3) at random times with respect to gamma phase or amplitude (Random treatment), or (4) during elevated mid-gamma at the same phases, but with the light path blocked at the entrance to the implanted ferrule (Sham treatment). The Random pulse delivery

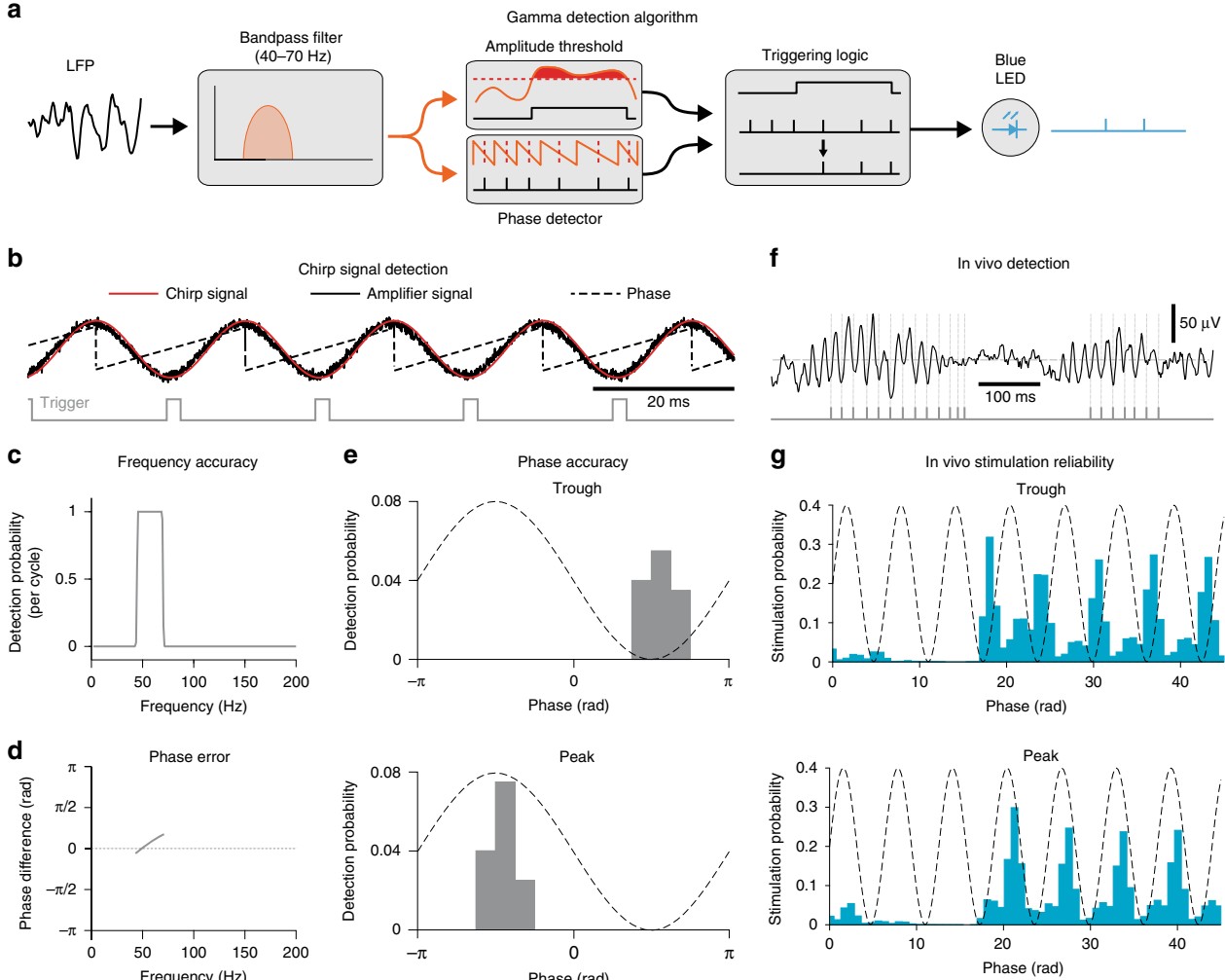

**Fig. 3** Description of gamma detection method and accuracy assessment. **a** Gamma detection method. **b** Example of trough detection (gray) on a chirp signal (red), amplified by our recording system (solid black). Dashed line: phase. **c** Chirp signal triggering probability across frequencies (selected band: 40–70 Hz). **d** Phase detection error for same signal. **e** Triggering probability distribution for Trough (top) and Peak detection (bottom). **f** Example of mid-gamma detection (gray) on LFP data. **g** In vivo stimulation reliability for trough (top) and peak (bottom, Trough: $n = 21$ sessions, 8 rats, Peak: $n = 25$ sessions, 9 rats)

control tested for the effects of optogenetic stimulation on the BLA irrespective of gamma, while the Sham control accounted for sensory cues caused by light stimuli. Every experimental day, only one type of modulation was used, randomized across animals and sessions, without a block structure. After one hour of stimulation, animals were taken to the HB apparatus for memory retention testing (Trials 2–4; Fig. 6a).

To ensure that the various groups were comparable in every possible way except for the gamma phase and amplitude dependence of the light stimuli, the following precautions were taken. First, the light intensity was adjusted so that 2 ms blue light pulses elicited a field potential of ~20 μV (Fig. 5j). Second, the amplitude threshold for gamma detection was adjusted separately each day such that the trigger frequency was ~20 Hz in the home box (Light OFF; Fig. 6b, c). Then, the same threshold was used in the subsequent Light ON session. Therefore, any increases or decreases in triggering frequency observed in the Light ON condition (Fig. 6b, c) were not due to changes in the modulation parameters but due to the combined influence of spontaneous changes in gamma occurrence and the effects of optogenetic light stimuli on gamma. Finally, to ensure that our algorithm would not trigger on the light-evoked field potential, we enforced a 10

ms minimum inter-pulse interval. Several factors indicated that our different modulation protocols were in fact similar. These included a stable stimulation frequency over time (Fig. 6c), similar inter-light stimulus interval (ISI) distributions (Fig. 6d) and an identical relationship between trigger rate and mid-gamma amplitude in all gamma-tracking groups (Trough, Peak, Sham; Fig. 6e). Thus, any differential effects observed on gamma power and behavior should be solely due to the phase dependence of light delivery.

Comparing the average gamma-band filtered BLA LFP around light pulses revealed that the Trough treatment enhanced mid-gamma amplitude, whereas the Peak treatment diminished it (Fig. 7a). This was also evident in the spectrograms (Fig. 7b) and power spectra reflecting the average changes between sessions (Light ON minus OFF; Fig. 7c). On the other hand, the Random treatment did not affect gamma levels (Fig. 7b, c). Since gamma oscillations tend to occur as brief bursts, we examined the effects of our stimulation protocol at the level of individual mid-gamma bursts (Fig. 7d). The Trough treatment increased the length and strength of gamma bursts (Fig. 7e, f), whereas the Peak treatment decreased burst amplitudes (Fig. 7f). Importantly, time spent in different sleep-wake states during the Light ON period did not

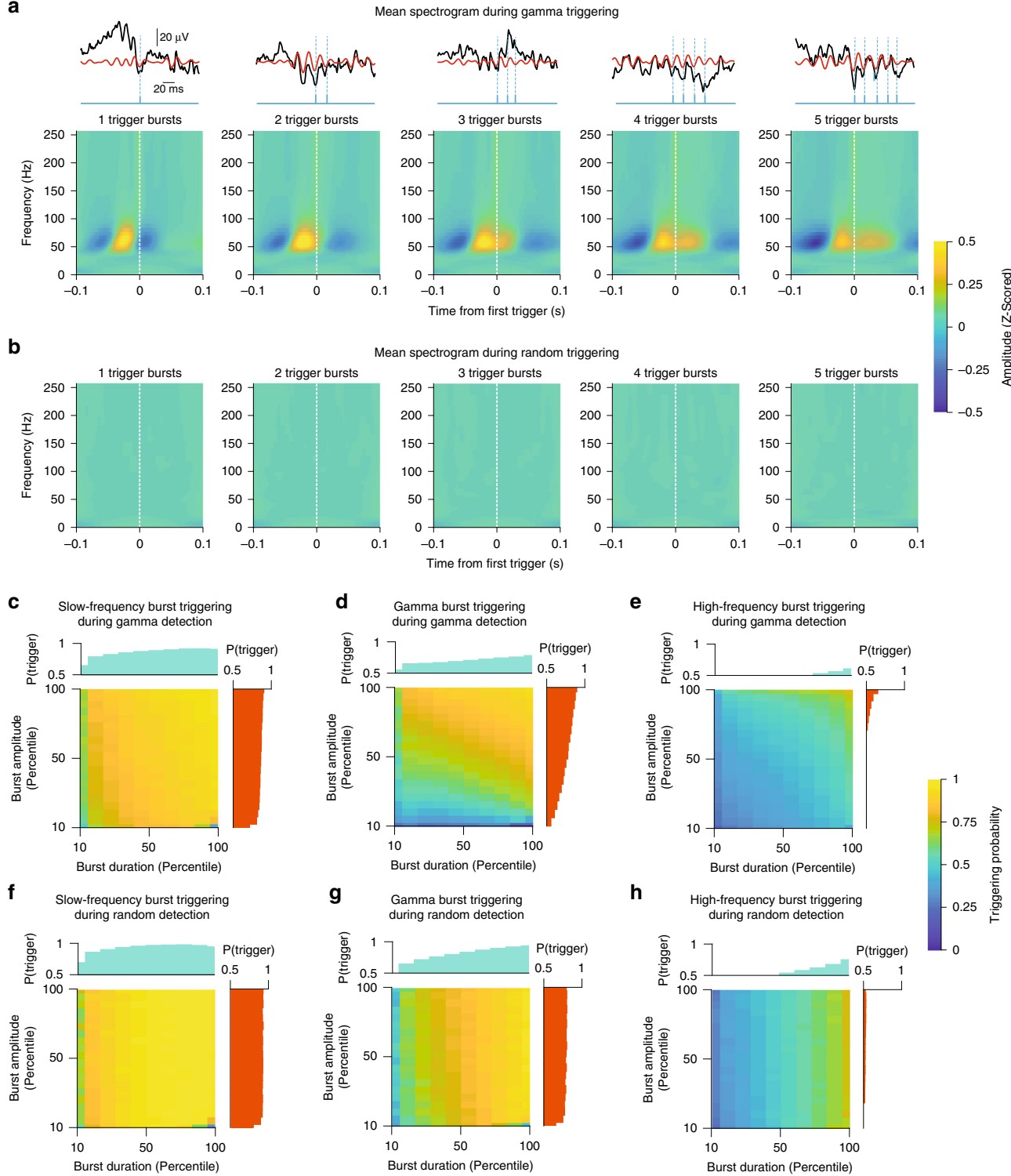

**Fig. 4** Gamma detection method shows frequency specificity and tracks the strength and duration of mid-gamma events. **a** Average BLA spectrograms around the first trigger generated by the gamma detection algorithm, for bursts containing different numbers of triggers (1-trigger to 5-trigger bursts, left to right). The first trigger of each burst is preceded by two above-threshold gamma cycles. Color bar: Z-scored spectral amplitude. Top: Example raw (black) and filtered (red) LFP traces for each burst size, along with trigger times (blue). **b** Similar spectrograms but during random triggering. **c** Probability of triggering within *slow-frequency bursts* (1–30 Hz) during *mid-gamma* detection, stratified by slow-frequency burst duration (*x*-axis) and amplitude (*y*-axis). Side bar graphs indicate marginal distributions for the corresponding dimension. Color bar: Triggering probability. **d** Probability of triggering within *mid-gamma frequency bursts* (40–70 Hz) during *mid-gamma* detection. **e** Probability of triggering within *high-frequency bursts* (90–250 Hz) during *mid-gamma* detection. **f** Probability of triggering within *slow-frequency bursts* (1–30 Hz) during *random* detection. **g** Probability of triggering within *mid-gamma frequency bursts* (40–70 Hz) during *random* detection. **h** Probability of triggering within *high-frequency bursts* (90–250 Hz) during *random* detection

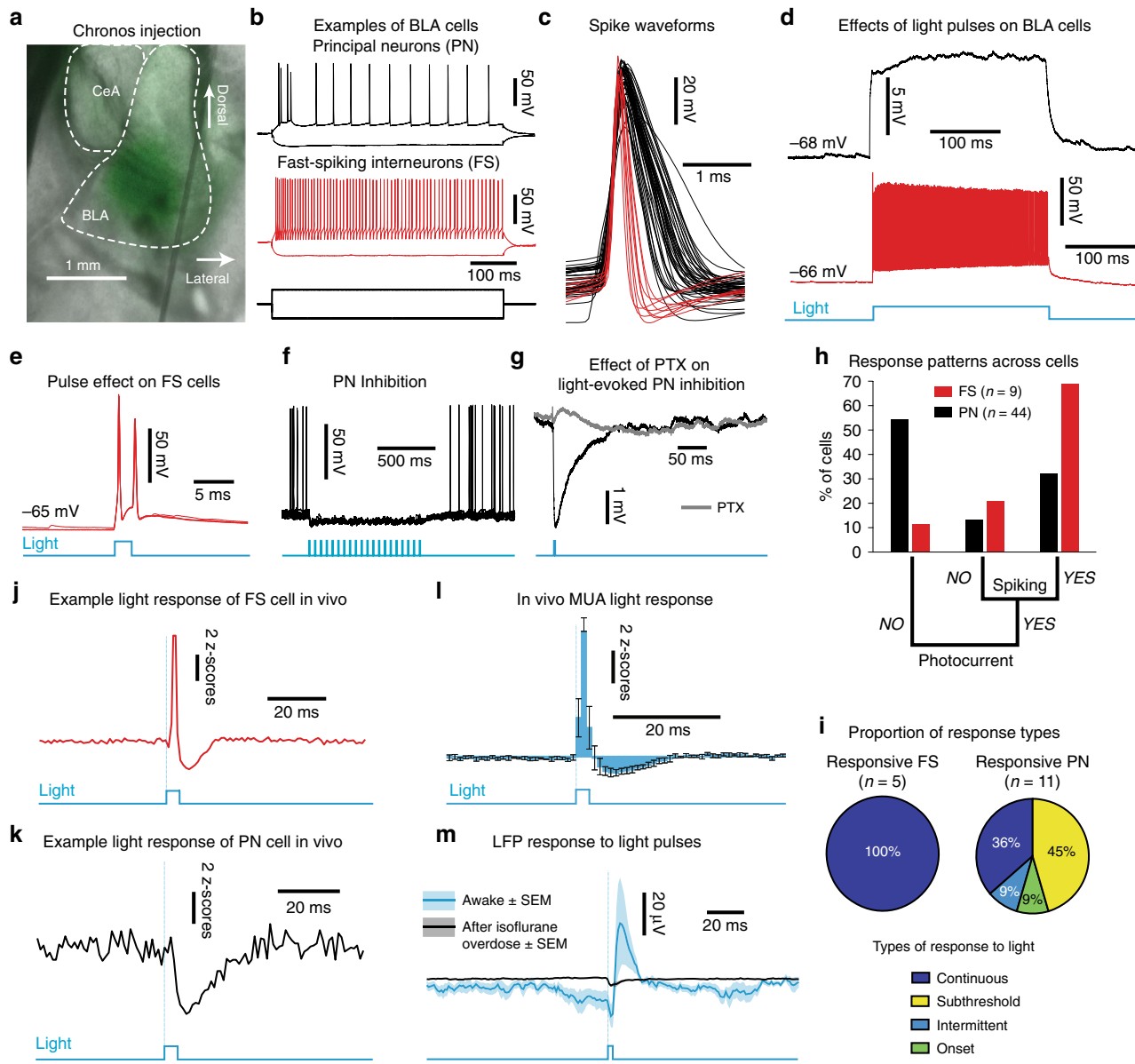

**Fig. 5** In vitro validation of optogenetic strategy. **a** Chronos injection (green) in the BLA. **b** Example recordings of BLA principal neurons (PN, black) and fast-spiking interneurons (FS, red). **c** Spike waveforms ($n = 53$ cells, 10 rats). **d** Example responses to a 500-ms light pulse (blue). **e** FS responses to 2-ms light pulses (blue). **f** PN inhibition by a light-pulse train (blue). **g** Effect of picrotoxin (PTX-100 µM, gray) on light-evoked inhibition of PN (black). **h** Cell response patterns (PN-black, FS-red). **i** Proportion of in vitro spiking at 50 Hz light stimulation (top) along with prevalent response types (bottom). **j** Example response of FS cell to light pulses in vivo. **k** Same for example PN cell. **l** In vivo multiunit activity (MUA) response to 2-ms light pulses ($n = 6$ sessions, 3 rats). Error bars: SEM. **m** Light-evoked LFP potential (average, blue ± SEM, shading) by 2-ms pulses is abolished by isoflurane overdose (black, $n = 5$ rats, Mann Whitney U(4) = 40, $p = 0.0079$)

vary between treatment types, indicating that they did not act by differentially altering behavioral states (Supplementary Fig. 7a).

We next examined how gamma modulation affects multi-unit entrainment in the BLA. All treatment types where light pulses were delivered caused a minor (~20%) firing rate decrease (Fig. 7i), consistent with the predominant recruitment of inhibition evident in our in vitro results (Fig. 5d, f, g). However, only the Trough and Peak treatments altered MUA phase locking to mid-gamma. Indeed, both treatments increased entrainment, but only when pulses were delivered during an individual burst (Fig. 7g, h; blue shading). Importantly, preferred phases differed between the Trough and Peak treatments, with the former augmenting entrainment to the preferred phase[13] and the latter shifting the

preferred phase by ~90 degrees backward, thus making neurons fire earlier than normal in the gamma cycle (Fig. 7j).

**BLA gamma modulation during consolidation affects memory.** Finally, we examined how modulating BLA gamma affects memory consolidation in the HB task by comparing latencies to the correct well as a function of treatment type (Fig. 8a). Note that no optogenetic stimulation was delivered during the retention phase. To assess mnemonic effects, we focused on the first retention trial (Trial 2), since every subsequent trial could depend in part on further training from the previous trial. For Trial 2, the Trough treatment reduced latencies to the

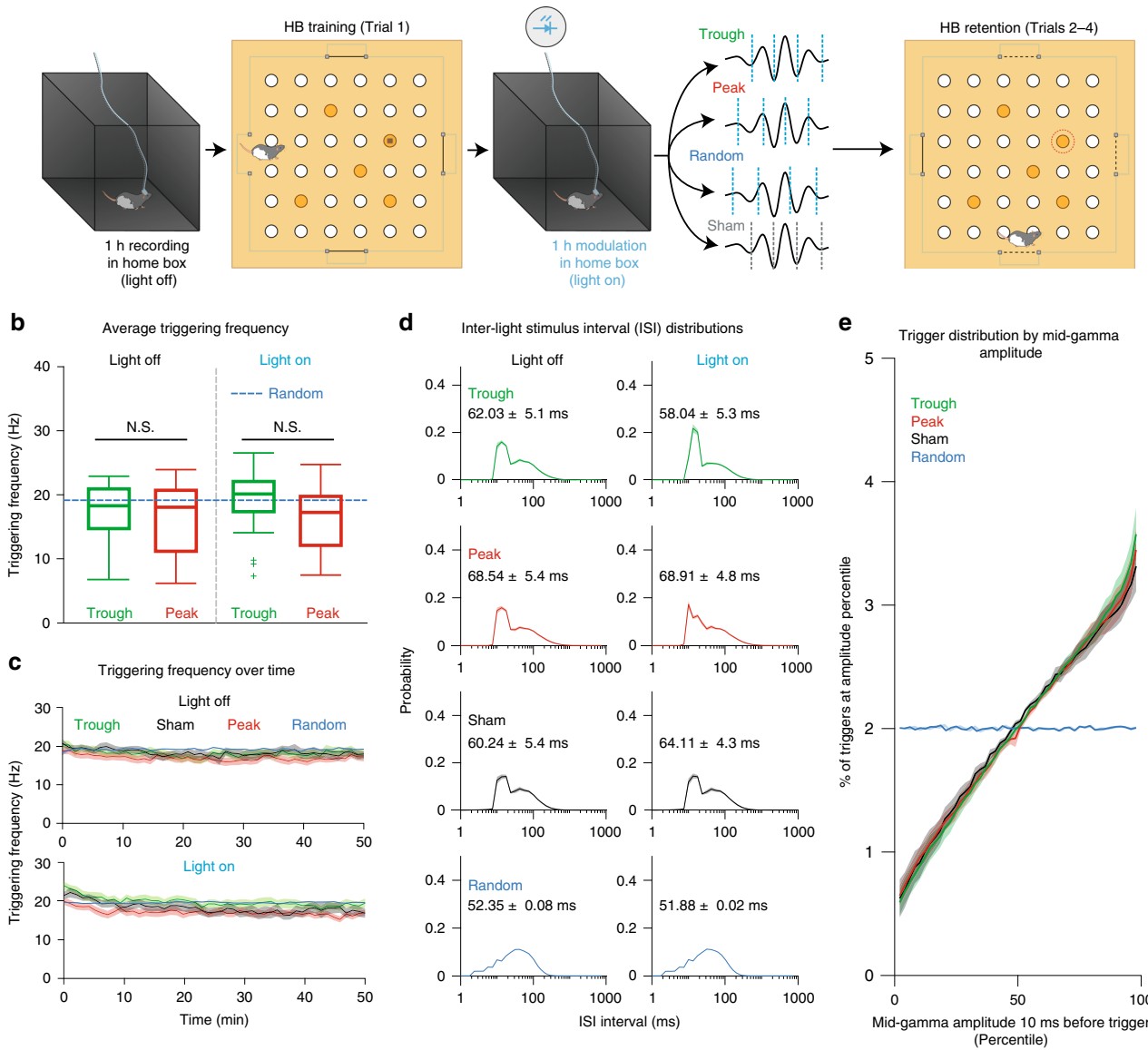

**Fig. 6** Types of gamma modulation and their stimulation properties. **a** HB training and modulation protocols. **b** Average triggering frequency for Light OFF (left) and Light ON (right) periods across treatments (Trough-green, Peak-red, Random-blue dashed line). The frequency of light stimuli in the Trough and Peak treatments were determined by the amplitude threshold used and gamma levels throughout the session. For Random, overall frequency was fixed at ~20 Hz. (Trough: $n = 21$ sessions, 8 rats, Peak: $n = 25$ sessions, 9 rats, Light OFF: Kruskal Wallis ANOVA $\chi^2(3) = 0.53$, $p = 0.77$, Light ON: Kruskal Wallis ANOVA $\chi^2(3) = 4.49$, $p = 0.10$). **c** Triggering frequencies calculated per minute across the entire session for all treatments. Top: Light OFF (Repeated Measures ANOVA $F_{(3,147)} = 1.36$, $p = 0.25$). Bottom: Light ON (Repeated Measures ANOVA $F_{(3,147)} = 2.13$, $p = 0.09$). **d** Average inter-light stimulus interval (ISI) distributions for all treatments. Left column: Light OFF, Right column, Light ON. Numbers in each ISI histogram correspond to average ISI ± SEM for each treatment. **e** Distribution of triggers depending on mid-gamma amplitude 10 ms before the trigger. Amplitude is expressed in percentiles based on the entire session. Blue: Random triggering, Green: Trough triggering at mid-gamma, Red: Peak triggering at mid-gamma, Black: Sham triggering at mid-gamma.

correct well compared to the Sham treatment (Fig. 8b). In contrast, the Peak treatment impaired retrieval, whereas Random light stimuli had insignificant effects (Fig. 8b). These results were not due to treatment-related differences in distances to the baited wells (Supplementary Fig. 7b, c). Interestingly, our gamma modulation protocol expanded the range of gamma levels present during the consolidation period (compare Fig. 8c to Fig. 1m). This unmasked a correlation between pre-to-post training changes in mid-gamma and performance in the HB task (Fig. 8c).

## Discussion

Our results indicate that gamma oscillations are enhanced in the BLA following an emotionally arousing experience, and that the strength of this enhancement correlates with subsequent memory. Moving beyond correlation, we experimentally tested the functional importance of these changes in gamma. By boosting or diminishing post-training BLA gamma, we bidirectionally modulated performance during a subsequent memory retention test. This suggests a causal relationship between BLA gamma and memory consolidation.

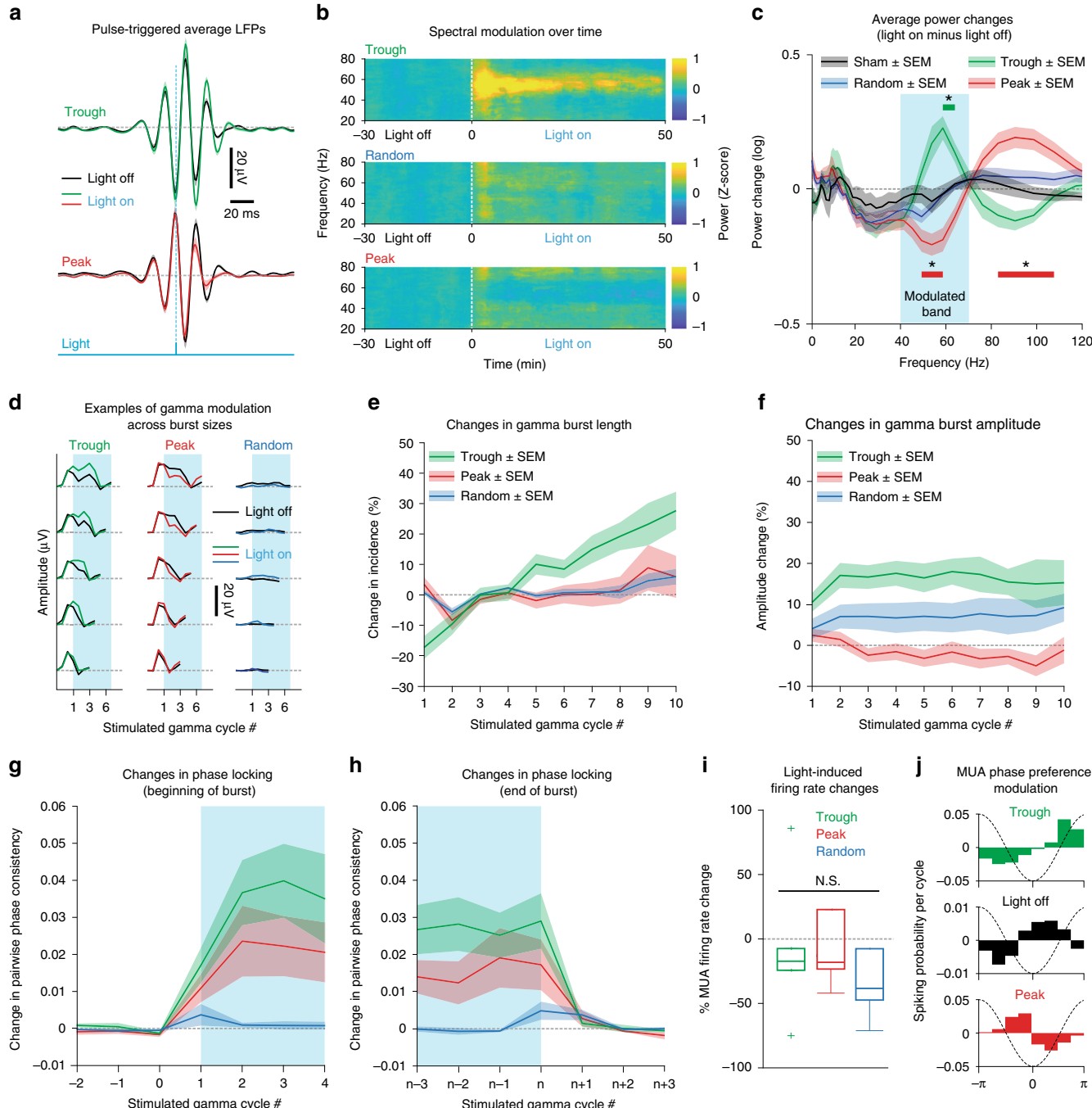

**Fig. 7** Real-time gamma modulation affects spectral power and unit entrainment. **a** Average filtered LFP around light pulses during Light OFF and ON. **b** BLA spectrograms for three modulation protocols. Dashed white line indicates Light ON start. **c** Average power changes (Light ON minus OFF). Blue rectangle outlines mid-gamma. Horizontal color lines indicate locations where a significant deviation from zero occurs per modulation condition (Kruskal Wallis ANOVA, $\chi^2(3) = 11.23$, $p = 0.01$; Trough: $p = 0.016$; Peak: $p = 0.039$ and $p = 0.021$, respectively. All $p$-values are Bonferroni corrected for multiple comparisons). **d** Gamma amplitude examples across burst lengths (Light OFF-black, ON-colored). **e** Changes in burst length (Kruskal Wallis ANOVA, $\chi^2(2) = 18.13$, $p = 0.0001$; Fisher's LSD post-hoc: Trough-Peak $p = 0.0003$, Trough-Random $p = 0.0001$, Peak-Random $p = 0.76$). **f** Burst amplitude changes (Kruskal Wallis ANOVA, $\chi^2(2) = 9.33$, $p = 0.0094$; Fisher's LSD post-hoc: Trough-Peak $p = 0.0025$, Trough-Random $p = 0.26$, Peak-Random $p = 0.064$). **g** Pairwise phase consistency (PPC) changes (Light ON minus OFF) before and after modulation starts in a gamma burst (Kruskal Wallis ANOVA, $\chi^2(2) = 10.35$, $p = 0.0057$; Fisher's LSD post-hoc: Trough-Peak $p = 0.26$, Trough-Random $p = 0.001$, Peak-Random $p = 0.034$). **h** Same for modulation end (Kruskal Wallis ANOVA, $\chi^2(2) = 9.02$, $p = 0.011$; Fisher's LSD post-hoc: Trough-Peak $p = 0.19$, Trough-Random $p = 0.003$, Peak-Random $p = 0.07$). **i** MUA % firing rate change (Light ON–Light OFF/Light OFF); Trough, $n = 6$ sessions, 3 rats; Peak, $n = 6$ sessions, 3 rats; Random, $n = 6$ sessions, 3 rats; Kruskal Wallis ANOVA $\chi^2(2) = 0.41$, $p = 0.815$. **j** Phase preference (Green: Trough, Black: All Light OFF recordings, Red: Peak. Exact $n$'s can be found in Supplementary Table 1

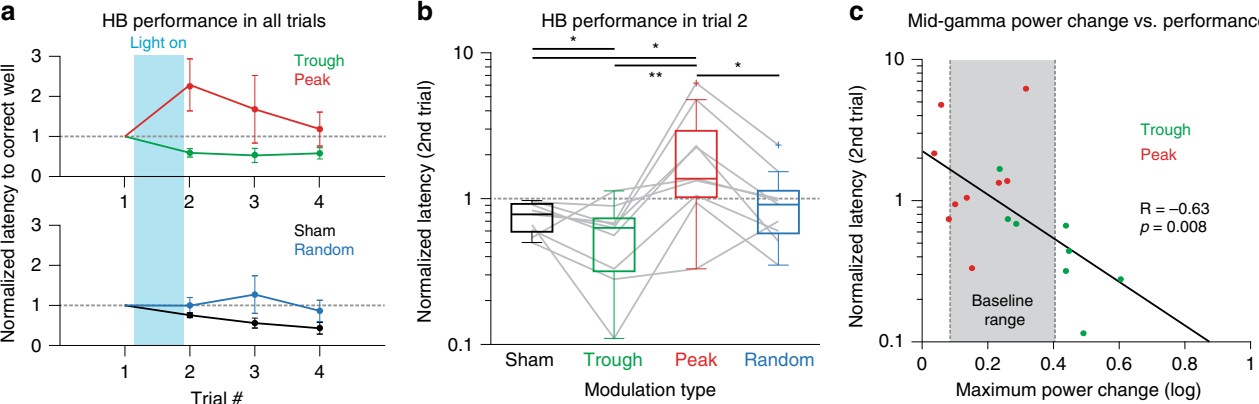

**Fig. 8** Real-time gamma modulation bidirectionally affects consolidation of appetitive spatial memories. **a** Normalized performance on all HB trials, for experimental treatments (top) and control treatments (bottom). Light blue box shows modulation window (hour between the first and second trial-Light ON). Trial 1 to Trial 2: 1 h ITI, Trial 2 to Trial 3: 5 min ITI, Trial 3 to Trial 4: 5 min ITI. **b** Normalized latency in the 2nd trial. Boxes show lower quartile, median and upper quartile; whiskers show the lowest and highest non-outlier observations. Gray lines show individual subject performance across treatment types (Kruskal Wallis ANOVA, $\chi^2(3) = 11.80$, $p = 0.0102$; Sign test post-hoc: Sham-Trough $p = 0.0391$, Sham-Peak $p = 0.0391$, Sham-Random $p = 1$, Trough-Peak $p = 0.0039$, Trough-Random $p = 0.17$, Peak-Random $p = 0.0391$). **c** Correlation between maximum mid-gamma power difference from zero (Light ON minus Light OFF) and normalized 2nd trial latency for Peak (red) and Trough (green) sessions. Each dot is the average of all same trial types for one subject. Solid line: least squares fit (Spearman's rank-order correlation $R = -0.6335$, $p = 0.008$). Combining performance from Peak and Trough groups shows expanded range of gamma levels caused by modulation. Gray box shows baseline gamma power range from Fig. 1m. Exact n's for every panel can be found in Supplementary Table 1

Systems consolidation is thought to depend on effective information transmission between regions[33]. Gamma oscillations have been implicated in such interactions because they appear in key regions for memory consolidation, such as the hippocampus and neocortex[34,35]. Importantly, hippocampal population events such as sharp wave ripples (SWRs), which are associated with memory consolidation[36], co-occur with bursts of gamma oscillations across many neocortical areas[37].

Why are gamma oscillations unique among neuronal rhythms in mediating these interactions? It was theorized that gamma has specific properties that help promote effective interregional communication[2–4]. Their frequency matches the timing of synaptic interactions between principal cells and interneurons, while also creating favorable circumstances for the induction of many types of synaptic plasticity[35,38,39]. Furthermore, they are frequently coupled with slower rhythms such as theta oscillations, which can facilitate the long-range routing of information between brain regions[40,41].

Since our results indicate that gamma oscillations are present in the BLA during consolidation, a question that arises is how can this BLA rhythm affect consolidation occurring elsewhere? Lesion and inactivation studies have shown that BLA activity is important during the early stages of consolidation for emotional experiences. Its recruitment shortly after emotional experiences is thought to facilitate mnemonic processes in regions that support memory storage and retrieval[14]. Anatomically, the BLA is bidirectionally connected with areas that participate in consolidation, such as the hippocampus and prefrontal cortex[42,43]. More recently, it was shown that the BLA exhibits oscillatory interactions with consolidation-related areas during emotional tasks[5,7,9,44]. Importantly though, these interactions persist after emotional experiences, with a recent study showing coordinated reactivation of amygdala-hippocampus neuronal ensembles during post-learning SWRs[45]. Since gamma oscillations could coordinate cortico-hippocampal interactions, the BLA is ideally positioned to facilitate this communication by augmenting the entrainment of both structures to gamma. Thus, in our experiments, by boosting or disrupting BLA gamma, we potentially affected these interactions, bidirectionally altering emotional memory strength.

Besides providing direct evidence for the role of BLA gamma in memory consolidation, the present in vivo study is also the first to achieve bidirectional control over endogenous gamma by conditioning stimuli on momentary changes in network excitability. Indeed, previous optogenetic studies used fixed stimulation routines to upregulate gamma[6,10–12,46], without taking into account the transient, "bursty" nature of this rhythm, as well as its possibly meaningful variations in frequency[47]. Indeed, the transient nature of gamma bursts may reflect discrete periods of information processing[21,48]. Thus, tonically altering gamma may force the network into an abnormal state[49]. Furthermore, previous experiments could not account for non-specific effects, such as overall changes in neuronal excitability. Lastly, no studies tested the effects of specifically downregulating gamma oscillations and thus could not causally link gamma to specific behaviors and cognitive processes.

Our findings highlight the importance of delivering light stimuli at specific phases of the endogenous gamma rhythm. This experimental design allowed us to overcome previous limitations and control for confounding factors, since the only parameter that changed between experimental conditions was the precise timing of light delivery with respect to the ongoing gamma phase. Yet, this very slight temporal difference (~7 ms on average) between conditions resulted in opposite effects on gamma levels and behavior. Given the variety of cognitive functions that gamma is thought to support, the present study introduces a versatile method with many potential applications in neuroscience research.

## Methods

**Surgery.** Adult male Long Evans rats (300–500 g, Charles River Laboratories) were anesthetized with ~2% isoflurane (Henry Schein) and placed in a stereotaxic frame (Kopf Instruments). After the skull was exposed and cleaned, unilateral craniotomies were performed over the BLA (Right BLA: AP −2.5, ML +5, DV −7.5, from brain surface). Tungsten wire tetrode arrays or single wires (20–50 μm diameter, California Fine Wire) were either inserted and fixed in place or advanced towards their target using chronically implanted 3D-printed microdrives[50]. For optogenetic

experiments, an optrode was inserted instead of an electrode array. The optrode consisted of a single tungsten wire or tetrode attached to a 200 μm, 0.39 NA optic fiber, coupled with a 2.5 mm FC/PC ceramic ferrule (Thor Labs). The distance between the fiber and electrode tips was ~1.5 mm. Electromyographic (EMG) and electroencephalographic (EEG) activities were also recorded using stainless steel skull screws. For silicon probe recordings, a probe with an 8 × 8 array of recording sites was inserted into the BLA and oriented in the coronal plane. This probe had 8 shanks, with 8 recording sites per shank. Each shank was 200 μm apart, and recording sites were 200 μm apart along the shank. Recording site size was 15 μm in diameter. The total width of the recording area is 1400 μm and its height is 1400 μm. Electrodes, optrodes, skull screws and electrode interface boards (EIBs) were secured to the skull using dental cement (Teets Denture Material, Co-Oral-Ite). All procedures were approved by the Institutional Animal Care and Use Committee (IACUC) of Rutgers University, in compliance with all ethical regulations defined by the Guide for the Care and Use of Laboratory Animals.

**Inhibitory avoidance (IA) paradigm and recordings.** We recorded spontaneous BLA activity for one hour (Pre-Training), in a box to which the subjects were previously acclimated ("home box"). Then, rats underwent IA training. The subjects were placed in a rectangular alley (Fig. 1a) comprised of a brightly lit compartment (52 cm (L)−25 cm (W)−30 cm (H)) connected to a dark section with a shock grid floor (29 cm (L)−35 cm (W)−30 cm (H)). The subjects' latency to enter the dark compartment was measured. To be scored as an entry, all paws had to touch the shock grid. After entering the dark compartment, re-entry into the light compartment was blocked by a sliding door and an electric footshock was delivered (1 mA for 1 s, Coulbourn Instruments). Immediately following training, activity was recorded for another hour in the "home box" (Post-Training). Retention was tested 48 h later by placing the subjects in the light compartment and measuring the latency to enter the dark section, with a cutoff point of 300 s. During this recall test, no shock was administered. Performance was measured as the difference between retention and training latencies. Subjects were divided into Good Learners and Poor Learners based on a median split of their performance distribution.

**Hole-board foraging task (HB) paradigm and recordings.** The HB task was adapted from a previously described version[22] with some modifications. The arena was a 180 × 180 cm square table with 6 × 6 equidistant sand wells (10 cm diameter, 12 cm depth) forming a square grid (Fig. 1g). On all sides of the apparatus, there were salient visual cues and starting boxes (20 × 10 cm) with a sliding door leading to the maze (Fig. 1g). The four sides were named North, East, South and West. Animals were placed on food restriction (14 g of rat chow per day, maintaining at least ~85% of initial body weight) for the entire duration of the experiments. After a few days of restricted access to food, rats were habituated to retrieving buried food pellets from sand wells inside their home cages (Pre-Training stage 1). After 3 days of exposure, animals that did not retrieve the pellet within 5 min were excluded. The remaining animals were exposed to the HB arena and trained to forage (Pre-Training stage 2). During this stage, only one out of 36 wells was accessible, while the rest had a lid on top of them. A visible food pellet was placed on the surface of the open well. Animals received four trials each day. In every trial, rats were placed in a start box and the door slid open, giving them access to the arena for 5 min. Both the start box and baited food well differed on every trial. Latency to retrieve the pellet was noted.

Once rats acquired the concept of foraging in the maze, as evidenced by decreased latencies to retrieve the food pellet, they continued into Pre-Training stage 3, where the uncovered baited well remained the same for the four trials, except that on the last trial the pellet was buried. Rats were trained on this version until they showed short latencies (<2 min) to the correct well and retrieved the buried pellet on the last trial. Rats that did not show adequate performance were excluded from the next phase.

Once rats were fully acclimated to the procedure, they were introduced to the final experimental phase of the task (Fig. 1h). In this phase, 5 out of 36 holes were uncovered on every training day, with only one of them containing the food reward. Rats underwent four trials every day. On trial 1 (Training Trial), the pellet was visible on the surface of the sand well. On trials 2–4 (Retention Trials), the pellet was either buried or absent (Probe Trial). Probe Trials were restricted to the third or fourth trial, so that rats could not predict when they would take place. Various precautions were taken to make sure that rats did not rely on any proximal cues, such as odor, to solve the problem: (1) mixing finely ground food into all sand wells, (2) using different start boxes on every trial, and (3) rotating the arena between trials while keeping the position of the baited well constant with respect to the distal cues. Furthermore, the outside rows of sand wells were not used, because their proximity to the walls could encourage thigmotaxis. Various Training-Retention intervals were tested (Fig. 1j), but the interval used for all modulation trials was 1 h. Latency to approach the correct well was measured on all trials and was normalized with respect to the first trial performance (Latency on trial N/Latency on trial 1).

**Endogenous activity recordings.** Two additional rats with a silicon probe implant were recorded while in the same home box as used for the HB foraging task. This

was a square (2 × 2 × 2 ft), black, enclosure with paper towel bedding on the floor, and dim room illumination. Recording sessions lasted for ~4 h.

**Virus injection.** Optogenetic control of neural activity was achieved with an adeno-associated viral vector (AAV) containing the Chronos channelrhodopsin variant[28], under a Synapsin promoter (AAV5.hSyn.Chronos-GFP, UNC Vector Core). Virus infusions (200 nl) were performed in the BLA using a pressure injector (Nanoject III, Drummond Scientific Company) and glass micropipettes (Narishige). After waiting for ~2 months to allow for full expression, implant surgeries were performed (see Surgery). During this time, animals were pre-trained in the HB task. On a subset of animals, in vitro experiments were performed (see next section).

**In vitro BLA slice recordings.** To characterize and validate the efficacy of Chronos, we obtained whole-cell patch recordings from BLA neurons. Rats were sacrificed with an overdose of isoflurane. Their chest cavity was opened, and they were perfused through the heart with a modified cold artificial cerebrospinal fluid (aCSF) for 30 s. The modified aCSF was comprised of (mM): 103 N-methyl-D-glucosamine, 2.5 KCl, 1.2 NaH₂PO₄, 30 NaHCO₃, 10 MgSO₄, 25 glucose, 20 HEPES, 101 HCl, 2 Thiourea, 3 Na-Pyruvate, 12 N-acetyl-L-cysteine, and 0.5 CaCl₂. Rats were then decapitated and their brain quickly removed, blocked, and sliced with a vibrating microtome (Dosaka) at a thickness of 300–400 μm starting at the anterior pole of the BLA. The cutting solution was the same cold modified aCSF mentioned above. Slices were then transferred to a chamber containing modified aCSF at 32 °C for 5 min, after which they were moved to a holding chamber at room temperature (22 °C) with normal aCSF (mM, 124 NaCl, 2.5 KCl, 1.25 NaH₂PO₄, 26 NaHCO₃, 1 MgCl₂, 2 CaCl₂, and 10 glucose, pH 7.2–7.3, 305 mOsm). Slices were kept in the holding chamber for at least an hour prior to recording.

To obtain whole-cell patch recordings, a slice was transferred to the recording chamber where aCSF heated to 32 °C continuously flowed. Slices were immobilized under a nylon net. A patch pipette (resistance 5–8 MΩ) filled with intracellular solution (mM: 130 K-gluconate, 10 N-2-hydroxyethylpiperazine-N'−2'-ethanesulfonic acid, 10 KCl, 2 MgCl₂, 2 ATP-Mg, and 0.2 GTP-tris(hydroxymethyl)aminomethane, pH 7.2, 280 mOsm) was guided towards a neuron with infrared video microscopy at 60× (Axioskop, Zeiss). We did not compensate for the liquid junction potential, which for this solution is 10 mV. We recorded from neurons in current-clamp mode using a Multiclamp 700B amplifier and digitized the signal with a Digidata 1550 (Molecular Devices).

Once the membrane potential stabilized, we delivered a graded series of 500 ms current pulses ranging from −200–360 pA in 40 pA steps. The first action potential evoked by the largest current pulse was used to classify neurons as either PN or FS. Neurons with action potential halfwidths <0.35 ms were classified as FS.

To assess optogenetic responsiveness, blue light stimuli were delivered for either 2, 5, or 500 ms. These could occur as either single pulses (2–500 ms) or trains of stimuli (5 ms pulses at 8 or 20 Hz for 1 s; 2 ms pulses at 50 Hz for 1 s). When classifying the optogenetic responsiveness of neurons, if a cell consistently emitted an action potential in response to any optogenetic stimulus, it was classified as spiking. If the neuron exhibited a rapid and sustained depolarization from rest, but no spiking, in response to the 500 ms current pulse, it was classified as having just a photocurrent. In some of the neurons that exhibited negative deflections in membrane potential upon delivery of blue light, we tested if it was mediated by GABAergic synapses by washing in the GABAₐ channel blocker picrotoxin (100 μM).

**Processing stages for closed-loop controller.** The closed-loop control algorithm was programmed in LabView and implemented on a USB-7845 reconfigurable multifunction data acquisition module (National Instruments). A Kintex-7 70 T field programmable gate array (FPGA) on the device instantiated the algorithm. Because the FPGA generates dedicated circuits for each processing step, the algorithm can run as fast as the clock rate of the chip, which is 80 MHz.

1. Analog to digital conversion: Signal is acquired as a fixed-point number (27 bits for digits, 5 bits for the scaling factor) and converted to a 32-bit signed integer to lower the logic overhead associated with numerical operations.

2. Clocking the algorithm: Processing goes in 40 μs steps, yielding an updating frequency of 25 kHz.

3. Obtaining the amplitude and phase: This module implements a bank of filters followed by control circuitry. The signal is passed separately to each filter with center frequencies at 35, 55, 75, and 95 Hz. For the purpose of our experiments, the only filter used was the one with 55 Hz as its center frequency. Within a band, two different bandpass filters are used:

a. To detect the phase, the signal is bandpass-filtered with a Butterworth filter (order 2, cutoff frequency g ± 15 Hz). If the filtered signal crosses from positive to negative, it is denoted as the descending phase, or if the filtered signal goes from negative to positive, it is denoted as the ascending phase. To detect peaks and troughs, we take the derivative of the filtered signal (with a lag of 2 ms between samples to reduce spurious detections). If this signal crosses from positive to negative a peak is registered, while if it crosses from negative to positive a trough is noted.

b. To detect amplitude, the signal is bandpass-filtered with a Butterworth filter (order 4, cutoff frequency g ± 15 Hz) and the absolute value of the filtered signal is taken. When either a trough or a peak is detected, then the amplitude value is updated.

A lower-order filter was used for phase detection to decrease phase distortion. A higher-order filter was used for amplitude detection to increase the frequency specificity of our estimated power in a particular band.

4. Restricting delivery to gamma bursts: To restrict the delivery of light-triggering pulses to gamma bursts, we set an amplitude threshold and a minimum number of cycles that needed to exceed that threshold. This was true for all conditions where gamma levels were tracked (Trough, Peak, Sham, also see Light Delivery section). Each day the amplitude threshold was set during the Light OFF condition so that on average pulses were delivered at ~20 Hz (Fig. 6b, c). In order for the triggering to start during a gamma burst, at least 2 cycles had to occur above threshold. We also could specify a maximum number of pulses to deliver per burst (for instance, to restrict delivery to 1 pulse per gamma burst), but left this value at 1000, which is effectively no maximum.

5. Light delivery: Triggering pulses were passed to a module that generated a 2 ms long digital output to drive a blue LED. A minimum interval of 10 ms was imposed between pulses, to avoid delivering more than one pulse per gamma cycle or retriggering on the light-evoked LFP response. Pulses were delivered at the trough (Trough treatment), peak (Peak treatment), randomly with an average frequency of 20 Hz (Random treatment) or during gamma but with light delivery blocked at the fiber stub (Sham treatment). The light source for the optogenetic experiments was a PlexBright compact LED module with blue light (465 nm − 24.9 mW − 792 mW/mm$^2$, Plexon Inc.). The LED module was coupled to ferrules on the animal's head through a PlexBright Dual LED 16 Channel Commutator and an FC/PC patch cable (Plexon Inc.). Light intensity was adjusted in each animal so that 2 ms pulses evoked a positive field potential of ~30 µV. The light intensity at the fiber stub required to elicit this field response was ~2.6 mW.

**Validation of closed-loop accuracy**. A function generator (AFG1062, Tektronix) was used to produce a sinusoidal waveform with a continuously ascending frequency from 1–200 Hz, known as a chirp. We attenuated the amplitude of the signal to the range of our preamplifiers and used it to drive our closed-loop algorithm. By simultaneously recording the output of the function generator and the triggering of our algorithm, we could assess any lags or distortions introduced by the preamplifier, analog-to-digital converter, and the signal processor.

**Histology**. At the end of the experiments, electrolytic marking lesions (10 µA, 10 s, A-M Systems) were performed for subsequent histological verification of electrode placement. Under deep isoflurane anesthesia, rats were perfused through the heart and their brain was removed and placed in fixative (4% paraformaldehyde in PBS, Sigma Aldrich). Two days later, the brain was transferred to a 30% sucrose solution (Sigma Aldrich). A few days later, the brain was sectioned on a freezing microtome (Reichert, 80 µm slice thickness). Sections were mounted on gelatin-coated slides and stained with thionin for determination of optic fiber and electrode placement. Some sections were left unstained and coverslipped to verify viral transfection on a confocal microscope (Fluoview FV1000, Olympus).

**Recording and data processing**. Unit activity and LFP were recorded with either a Plexon or an Intan recording system (30 kHz/channel sampling rate, Plexon Inc./Intan Technologies) through 16-channel or 96-channel EIBs (Neuralynx) on the animal's head. The resulting binary files were then processed offline using NDManager[51] and MATLAB (Mathworks).

Unit activity was detected and sorted based on the spike waveforms using KlustaKwik[52] and then manually clustered using Klusters[51]. For clustering, only the first four principal components were used for each electrode. A waveform cluster was considered a real unit if it had a refractory period in its auto-correlogram, as well as a characteristic waveform shape (negative followed by positive deflection) that did not resemble typical artifact/EMG shapes. Pooling all detected units together yielded multiunit activity (MUA). We restricted our MUA analyses to sessions that had a stable MUA presence throughout the session (>99%) and >2 Hz firing rate.

For LFP analyses, a Butterworth lowpass filter with a 300 Hz cutoff point (2nd order filter) was used in both forwards and backwards directions to avoid phase distortion. The LFP was then down sampled to a 1 kHz sampling rate. Recordings that had high EMG or electrical noise were excluded from spectral analyses.

For phase locking analyses (Fig. 7g, h), we calculated the pairwise phase consistency[53] (PPC) of the MUA to mid-gamma. The LFP was filtered between 40 Hz and 70 Hz only in the forward direction, to avoid contamination by the light-evoked LFP response. Phase information was extracted from the filtered trace and MUA phase locking was calculated. Change in PPC was calculated as the difference between Light ON and Light OFF PPC for any gamma cycle. Spiking probability per phase was calculated as the average firing rate per gamma cycle at any phase. Then, the mean spiking probability was subtracted from all phases to show deviation from the mean.

Frequency spectra were calculated using Welch's power spectral density estimate. A power law function was fitted to the raw spectrum and then subtracted from it to show residual power, accounting for overall changes in spectral power and individual differences (Supplementary Fig. 1a, b). Maximum changes in spectral power and corresponding frequencies (Supplementary Fig. 1f) were calculated by finding the local maximum in the difference spectra (Light ON minus OFF) between 40 Hz and 70 Hz.

Time-frequency spectrograms for Figs. 1d, k, 7b were calculated using the Chronux package[54] for MATLAB. Spectral power on the LFP time series was detected using a multi-taper technique (2nd order filter), for multiple estimates of power within a fixed frequency band and time window. In all spectral analyses, 5 tapers were used. The window size was 2 s, overlapping by 250 ms. For baseline IA and HB analyses, baseline was 30 min prior to training. For optogenetic experiments, baseline was the entire Light OFF period.

Time-frequency spectrograms, as well as average spectral amplitudes for Fig. 4a, b were calculated using Morlet wavelets ranging from 1 to 256 Hz in quarter octave steps. The width of the wavelet was seven cycles. To measure the amplitude at a particular frequency and time, we took the absolute value of the complex valued frequency domain representation of the signal.

For individual burst analyses in Supplementary Fig. 2, mid-gamma bursts were detected in the wavelet power spectrogram. Then, their individual durations and amplitudes were extracted, both before and after training in the IA and HB tasks. Changes in duration, amplitude and incidence were calculated between 30 min before and after training in both tasks.

For individual burst analyses in Fig. 7d–h, the LFP was filtered in the forward direction and burst cycle numbers and amplitudes were extracted during periods when light pulses were delivered. For each light-modulated burst, 3 cycles before and after the light delivery were also extracted to show specificity of light effects. Changes in amplitude and incidence were calculated based on differences from the Light OFF period.

For analyzing the synchrony of gamma bursts along the dorsoventral and mediolateral axis (Fig. 2d, e) we used data obtained from the subjects implanted with an 8 × 8 silicon probe in the BLA. Mid-gamma bursts were detected in the wavelet power spectrogram at a site midway along the shank. The time of each burst corresponded to the trough of the oscillation that was nearest to a local peak in the power of the spectrogram. To visualize synchrony in the gamma burst waveform across sites, for each site we obtained the mean LFP centered at those peak times with a window of ±75 ms. We also measured the distribution of phases in the wavelet spectrogram at each site by applying the angle function to the complex valued wavelet spectrogram at the peak times.

Recordings from the 8 × 8 silicon probes were also subjected to current source density (CSD) analysis to determine whether the sources underlying the LFP gamma observed in the BLA were local (Fig. 2f, g). Due to the non-laminar structure of the BLA, and the two-dimensional arrangement of recording sites, we used the inverse current source density technique to obtain estimates the CSD[55]. Unlike traditional approaches that take the second spatial derivative along a single spatial dimension, the inverse current source density method inverts the electrostatic equations that map current sources at the recording sites onto recorded potentials. This matrix can then be used to solve for the underlying CSD map. In constructing this matrix, we used spline smoothing to minimize spatial noise. Once the CSD had been calculated, we calculated its power spectrogram for each site and compared the mean spectrum between sites recorded in the BLA and those in adjacent central amygdala or striatum.

Sleep-wake state detection was conducted in a semi-automated manner using EMG and EEG activity along with video motion detection.

**Statistical analyses**. All statistical tests were conducted using MATLAB. For IA performance, a two-way analysis of variance (ANOVA) was performed with Group (two levels) and Session (two levels) as independent variables. We used a two-way ANOVA because there is no non-parametric equivalent for multi-factorial analyses. For HB performance, a Kruskal Wallis ANOVA was performed with normalized latency to the food reward as the independent variable. Unpaired post-hoc comparisons were performed using Fisher's Least Significance Difference (LSD), while paired comparisons were performed using the sign test. For average power changes between performance groups or types of modulation, a Kruskal Wallis ANOVA (for more than two levels), or a Mann Whitney test (for two levels) was conducted, with maximum power change as the independent variable. For assessment of the relationship between gamma power changes and performance, a Spearman's rank-order correlation was performed. For individual burst treatment differences, a Kruskal Wallis ANOVA was performed on the incidence or amplitude change from baseline. For comparing MUA changes or phase locking changes, a Kruskal Wallis ANOVA was performed on percent firing rate changes or PPC changes relative to baseline for light-modulated cycles. Finally, for comparing sleep-wake state durations, a Mann Whitney test or a Kruskal Wallis ANOVA was performed on the percent time spent in each state between groups. All statistical tests were two-tailed, and significance was judged at the 0.05 level.

**Reporting summary**. Further information on research design is available in the Nature Research Reporting Summary linked to this article.

## Data availability

The data that support the findings of this study are available from the corresponding author upon reasonable request.

## Code availability

The code that was used for closed-loop control is available on https://github.com/dbheadley/GammaGovernor

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

## Acknowledgements

This work was supported by R01 grants MH107239 and MH112505 to D.P. from NIMH. We thank the Behavioral and Neural Sciences Graduate Program for the support of V.K.

## Author contributions

V.K. performed in vivo experiments, analyzed data, contributed to experimental design and wrote the first draft of the paper. D.P. conceived the adapted version of the Hole-board Task, helped with figure design and co-supervised the project. D.B.H. designed and developed the closed-loop method, performed in vitro experiments, analyzed data, contributed to experimental design, and co-supervised the project. V.K., D.P. and D.B.H. revised and edited the paper.

## Additional information

**Competing interests:** The authors declare having no competing interests.

