## [Peer Review File · Nature Communications]

Reviewers' comments:

Reviewer #1 (Remarks to the Author):

This paper describes a novel method for phase-specific manipulation of gamma oscillations and its application to the basolateral amygdala during a memory-guided foraging task. The intervention is remarkably precise compared to previous attempts to manipulate rhythmic activity in the brain, which either stimulated gamma in open-loop mode, or performed closed-loop stimulation of slower oscillations, such as theta. The fact that the authors are able to bidirectionally modulate task performance within individual subjects using only unilateral stimulation suggests that the effect of their intervention is quite powerful. The title undersells its impact, as even a negative result could be considered a "test" of gamma's role in cognitive processes. Overall, this is a well-executed study with interesting findings, but the authors should address the following comments prior to publication:

Figure 1:

It is well-documented that gamma occurs as brief bursts, and the authors' closed-loop intervention is optimized to deal with gamma power changes on a rapid timescale. However, Figure 1 focuses on gamma fluctuations measured via average power spectra over long intervals, which are an indirect measure of gamma burst occurrence. The correlation between behavioral performance and BLA gamma may strengthen when looking at a more direct measure of gamma bursts, such as the changes in burst length and burst amplitude. Also, more information should be provided in supplementary Figure 1, which shows potentially interesting differences in power spectra across subjects. Some display a sharp peak around 50 Hz, while others have a broad peak between 50 and 100 Hz, while others show both. Identifying the subjects from which these spectra originated could indicate whether there are differences in the overall patterns of gamma activity expressed in the IA vs HB task, or good or poor learners, etc.

Figure 2:

The authors use current source density analysis to demonstrate that gamma oscillations are localized to the BLA. This is the correct approach, but the results shown in Figure 2 are not very convincing. The overall shape of the power spectrum is quite similar between BLA and adjacent areas, such that overall power differences could be explained by differences in cell density, for example. With data from an 8 x 8 electrode array, the authors have the opportunity to make a more compelling case for gamma localization in BLA, perhaps with burst-triggered CSD plots from multiple electrode columns, or by showing a heatmap of gamma burst rate across all recording sites. Again, focusing on gamma bursts, rather than changes in the average power spectrum, is more relevant to the rest of the manuscript. Also, Figure 2c and d should indicate the number of bursts that went into the average, and what filtering was applied, if any. Overlaying an example of an individual gamma burst would be helpful.

Figure 3:

The authors do an excellent job illustrating how their algorithm works and validating its performance on artificial signals and in vivo. In addition, they should indicate how they plan to make their LabView code available to other researchers interested in performing similar manipulations.

Figure 4:

What is the meaning of a 1-trigger burst? Is it just a threshold crossing that lasts for a single cycle of gamma? It would be helpful to see a histogram of the overall number of triggers per burst. Figures 4c and 4f are a bit confusing—why is the probability of triggering so high for slow-frequency oscillations? Does this just reflect the fact that gamma bursts are often embedded within slower rhythms?

Figure 5:

If I understand Figure 5h correctly, it's indicating that FS cells display light-evoked spiking with a much higher probability than PNs, but PNs still spike roughly 30% of the time. If so, it would appear that the effect of stimulation is primarily excitatory, given the relative abundance of PNs in the BLA. This contrasts with the authors' statement that "Chronos activation [...] exerted predominantly inhibitory effects." Further analysis of the data that produced Figure 5i could shed some light on this. Even if well-isolated single units are not available, the authors could classify individual spike waveforms as fast-spiking or regular-spiking, and plot their probability of occurrence relative to the time of light onset.

Figure 6:

If the average triggering frequency is ~20 Hz, and gamma bursts are being filtered around 55 Hz, it means that stimulation is active about 1/3 of the time. How does this compare to the relative amount of time the amygdala spends in a putative gamma state under baseline conditions? Is this value consistent across subjects?

Figure 7:

The bidirectional change in gamma power between 40 and 70 Hz seems to be primarily mediated by changes in gamma burst amplitude. As stated previously, reporting similar metrics in Figure 1 would be beneficial.

Reviewer #2 (Remarks to the Author):

In this manuscript, Kanta et al., develop a method to accurately detect and modulate gamma oscillations, in freely moving rats. Using this method, they establish the causal relationship between post-learning baso-lateral amygdala (BLA) gamma oscillations and consecutive performance during memory retention test of a contextual appetitive task (referred by the authors as HB). The present data are good quality offering a new method for the community and knowledge about BLA physiology. However, I think that the authors could exploit more the data (these data are rare in BLA) and clarify some statements. Please find my comments organized as follow: general and figure-by-figure ones.

1/ The authors develop a method to accurately manipulate gamma oscillations. However, the authors use gamma power as unique measure to characterize gamma oscillations during behavior (in figure 1). The closed-loop method is doing more than affecting gamma power. The authors could establish the correlation between of mind-gamma events measure (number, strengths and durations) or other measures and consecutive performance during memory retention test, especially for HB task.

2/ The authors infer from the presence of gamma oscillations in BLA after aversive and appetitive trainings, that consolidation of both task share similar brain mechanism, but this is an over statement. Regretfully, they didn't manipulate gamma oscillations after contextual fear task training (referred by the authors as IA), despite the fact that BLA gamma oscillations after IA training are strong and correlate with consecutive performance during memory retention test. Is the presence of IA data in the paper necessary? Indeed, those data do not support the main conclusion of the paper regarding the role of BLA gamma during consolidation (since it has not been tested) and is not use to test the validity of the closed-loop method (as suggested by the manuscript title). An example of such confusing situation where the authors mix results obtain with IA or HB task, can be read in the lines 248 to 254 of the discussion section. The authors are not able to establish correlation between pre-to-post training changes in mid-gamma and consecutive performance for the HB task, but they are for the IA task. So how they could "experimentally tested the functional importance of these changes in gamma".

3/ This is a general comment for all figures containing electrophysiology. It would great to see raw

LFP (as it is generally done in papers from Pare's lab). The authors use raw LFP to show phase detection only. Also they use the same example for figures 3 and 6.

4/ Comment related to Figure 1:

Regarding IA, the authors related the presence of post-training BLA gamma oscillations to memory consolidation by citing previous studies (line 49, 4 studies, not really conclusive from my point of view), ignoring studies linking BLA gamma oscillations to fear expression (from Joshua Gordon, Cyril Herry and other labs). As mention earlier, without manipulation of gamma with the closed-loop method, the authors cannot determine the role of post-learning BLA gamma oscillations in IA task consolidation. One plausible explanation is that post-learning BLA gamma oscillations correlated with fear behavior. This could explain also the difference between poor and good learners. The authors should tackle the following question: What are the rats doing during high BLA gamma (that lasted around 20 min) after training? Instead of plotting the average % of time spend in the 3 different sleep-wake states in Supplementary figure 1c, the authors could show how the behavior (the 3 sleep-wake states for example) distribute along the post-training session, for IA and HB tasks.

4/ Comment related to Figure 2:

In this figure the authors claim that BLA gamma is locally generated. However the only evidence comes from the fact that gamma power is stronger in BLA than CEA and STR. This is not enough. Indeed, CEA and STR are not the only structures surrounding BLA from which gamma could come from. The authors could exploit more the 8x8 silicon probes used. I couldn't find any details about those probes and recordings, like: length of shanks, space between shanks and histological locations. The authors show phase-coherence along the dorso-ventral axis (is that along all 8 recording sites?) of BLA but not in the medio-lateral axis. What is the phase-coherence between BLA, CEA and STR? More importantly, the authors have to show that locally recorded BLA gamma entrain local units and not just cite previous work.

5/ Comment related to Figure 5:

I think that the goal of this figure should be to evaluate the effect of the stimulations used in vivo on BLA neurons. However the authors describe how neurons response to single light pulses, in vitro and in vivo. There is only one example of PN response for light pulse-train around 20hz, not gamma range! They use Chronos to maximize the temporal precision of the method, they should show it. Because of the model (rats) the authors use a non-specific optogenetic strategy, it's fine, but from this figure we don't now what is the activity of neurons during gamma modulations, in vitro and in vivo.

6/ Comment related to Figure 8:

This is the most important figure, validating the method. It's really interesting results, regretfully, we don't know if it's working for IA task, even with inter-subjects comparisons. 8c is confusing to me, why pulling together peak and trough groups for the correlation calculation? It makes more sense to separate the two groups. Peak group don't show correlation between gamma power and performance, as in figure 1, but the trough group show high correlation.

REVIEWER #1:

REVIEWER'S GENERAL COMMENTS: This paper describes a novel method for phase-specific manipulation of gamma oscillations and its application to the basolateral amygdala during a memory-guided foraging task. The intervention is remarkably precise compared to previous attempts to manipulate rhythmic activity in the brain, which either stimulated gamma in open-loop mode, or performed closed-loop stimulation of slower oscillations, such as theta. The fact that the authors are able to bidirectionally modulate task performance within individual subjects using only unilateral stimulation suggests that the effect of their intervention is quite powerful. The title undersells its impact, as even a negative result could be considered a "test" of gamma's role in cognitive processes. Overall, this is a well-executed study with interesting findings, but the authors should address the following comments prior to publication:

AUTHORS' RESPONSE: We thank the reviewer for the suggestions. In response to this overall comment, we changed the title of the manuscript from "*Closed-loop optogenetic control of gamma oscillations directly tests their role in cognitive processes*" to "*Closed-loop optogenetic control of gamma oscillations demonstrates their role in cognitive processes*"

REVIEWER COMMENT 1: Figure 1: It is well-documented that gamma occurs as brief bursts, and the authors' closed-loop intervention is optimized to deal with gamma power changes on a rapid timescale. However, Figure 1 focuses on gamma fluctuations measured via average power spectra over long intervals, which are an indirect measure of gamma burst occurrence. The correlation between behavioral performance and BLA gamma may strengthen when looking at a more direct measure of gamma bursts, such as the changes in burst length and burst amplitude. Also, more information should be provided in supplementary Figure 1, which shows potentially interesting differences in power spectra across subjects. Some display a sharp peak around 50 Hz, while others have a broad peak between 50 and 100 Hz, while others show both. Identifying the subjects from which these spectra originated could indicate whether there are differences in the overall patterns of gamma activity expressed in the IA vs HB task, or good or poor learners, etc.

AUTHORS' RESPONSE: As requested, we have analyzed the relation between behavioral performance and various properties of gamma bursts (duration, amplitude, incidence) in the IA and HB tasks. These results are shown in a new supplementary figure (**Supplementary Fig. 2** in the updated manuscript). Consistent with the much larger increase in gamma power in the IA than in the HB task, we found a significant correlation between behavioral performance and gamma burst incidence (but not duration or amplitude) in the IA task but not in the HB task. As to the reviewer's second request, **Supplementary Fig. 1f** shows the gamma frequency with the largest power increase in the IA (left) and HB (right) tasks. No obvious difference can be detected between the good and poor learners or between the two tasks. Similarly, no difference was found between the half-width of the frequency band showing the largest increase between the poor and good learners or between the two tasks (**Supplementary Fig. 1g**).

REVIEWER COMMENT 2: Figure 2: The authors use current source density analysis to demonstrate that gamma oscillations are localized to the BLA. This is the correct approach, but the results shown in Figure 2 are not very convincing. The overall shape of the power spectrum is quite similar between BLA and adjacent areas, such that overall power differences could be explained by differences in cell density, for example. With data from an 8 x 8 electrode array, the authors have the opportunity to make a more compelling case for gamma localization in BLA,

perhaps with burst-triggered CSD plots from multiple electrode columns, or by showing a heatmap of gamma burst rate across all recording sites. Again, focusing on gamma bursts, rather than changes in the average power spectrum, is more relevant to the rest of the manuscript. Also, Figure 2c and d should indicate the number of bursts that went into the average, and what filtering was applied, if any. Overlaying an example of an individual gamma burst would be helpful.

AUTHORS' RESPONSE: We agree with the reviewer that the CSD power spectra for BLA and CeA/STR sites were broadly similar in appearance, however this is not entirely unexpected, since virtually all local field potentials exhibit a $1/f$ frequency falloff (e.g. Miller, Sorensen, Ojemann, and Nijis 2009). This manifests as a linear decrease in power with increasing frequency when the power spectrum is plotted on a log-log plot, which ours is. Crucially, when we examined where these curves deviated the most, by taking the ratio of the power between the BLA and CeA/STR recording sites, we found that the strongest difference was in the gamma band, peaking at the gamma frequency we focused on in this manuscript (**Revised Figure 2g**). Furthermore, the fact remains that there is a bump in the gamma band for the BLA CSD power spectrum, which should principally reflect local transmembrane currents.

The occurrence of gamma oscillations in the BLA has been reported for almost two decades (Collins, Pelletier, and Pare 2001), and recently it has been found to correlate with various behavioral states (e.g. Amir, Headley, Lee, Haufler, and Pare 2018; Stujenske, Likhtik, Topiwala, and Gordon 2014). The study by Amir et al. (2018) also showed that PNs and FSs in the BLA fire during the phases of gamma that one would expect if they were involved in local generation of the rhythm via a Pyramidal-Interneuron Network Gamma mechanism (PING, Whittington, Traub, Kopell, Ermentrout, and Buhl 2000). Follow up on this work using a biophysically realistic model of the BLA demonstrated the sufficiency of its local circuitry to generate gamma that matches characteristics measured in the local field potential in vivo (Feng, Headley, Amir, Kanta, Chen, Pare, and Nair 2019). Thus, both the prior literature and the present results support the position that the BLA is capable of generating gamma oscillations.

However, we also thought the suggestions made by the reviewer regarding Figure 2 would enhance these points and provide a substantially richer picture of BLA gamma. To address the request that we examine gamma bursts more closely in the 8x8 silicon probe recordings, we ran the same burst analysis used in revised **Figure 2d,e** on the LFP from all recording sites from both subjects. Both the rate of burst occurrence and the mean burst amplitude were detected. Both the gamma burst rate and burst amplitude showed prominences within the BLA (**Figure 2b,c**). Gamma bursts occurred less frequently and with lower intensity in adjacent STR and CeA sites, regions that lack the local circuitry to support gamma via a PING mechanism. As a positive control, we also found robust gamma bursts in adjacent cortical regions (leftmost column and bottom row of **Figure 2b,c**), which agrees with the propensity for cortical circuits to produce gamma.

We also made the requested changes to the mean burst panels and the phase histograms. The number of bursts used to calculate these graphs is now included in the figure legend ($n=51964$), and it is noted that no filtering was used for the gamma burst waveform plots. We have also added a similar mean burst waveform / phase histogram plot for the medial-lateral axis.

To give readers a feel for the raw data arising from the silicon probe recordings, a supplementary figure is now included showing an example gamma burst recorded from all 8x8 sites on the silicon probe (**Supplementary Fig.a 3**).

REVIEWER COMMENT 3: Figure 3: The authors do an excellent job illustrating how their

algorithm works and validating its performance on artificial signals and in vivo. In addition, they should indicate how they plan to make their LabView code available to other researchers interested in performing similar manipulations.

AUTHORS' RESPONSE: We thank the reviewer for this suggestion. Upon acceptance of the paper, we plan to create a Github page that will contain the LabView code along with instructions for potential users.

REVIEWER COMMENT 4: Figure 4: What is the meaning of a 1-trigger burst? Is it just a threshold crossing that lasts for a single cycle of gamma? It would be helpful to see a histogram of the overall number of triggers per burst. Figures 4c and 4f are a bit confusing—why is the probability of triggering so high for slow-frequency oscillations? Does this just reflect the fact that gamma bursts are often embedded within slower rhythms?

AUTHORS' RESPONSE: As mentioned in the original manuscript, to avoid detection of spurious events that are not true gamma bursts, we included a restriction in our algorithm whereby two consecutive gamma cycles above threshold must be detected before light is delivered on the third (above-threshold) cycle. Thus, in this figure, a 1-trigger burst is one that had three cycles above threshold. The figure legend now clarifies this point.

As requested, the revised paper (revised **Supplementary Fig. 4**) now includes a heatmap showing the average number of triggers per burst, stratified by burst duration and amplitude.

The reviewer is correct about the explanation of 4c and 4f. Gamma bursts often co-occur with slow-frequency events, which elevates the overall probability that the slow frequency events contain a pulse. However, both figures show that the triggering probability does not change with the amplitude of slow rhythms, which indicates that our algorithm is not tracking these events instead of gamma.

REVIEWER COMMENT 5: Figure 5: If I understand Figure 5h correctly, it's indicating that FS cells display light-evoked spiking with a much higher probability than PNs, but PNs still spike roughly 30% of the time. If so, it would appear that the effect of stimulation is primarily excitatory, given the relative abundance of PNs in the BLA. This contrasts with the authors' statement that "Chronos activation [...] exerted predominantly inhibitory effects." Further analysis of the data that produced Figure 5i could shed some light on this. Even if well-isolated single units are not available, the authors could classify individual spike waveforms as fast-spiking or regular-spiking, and plot their probability of occurrence relative to the time of light onset.

AUTHORS' RESPONSE: In vitro physiological studies in the BLA have revealed that the incidence of connections from FS cells to PNs is much higher than of connections from PNs to FS cells (Woodruff and Sah, 2007). Consistent with these findings, we found that in response to blue light, the vast majority of FS cells fired and that most PNs were inhibited (only 30% of PNs fired). That's why we stated "*Chronos activation [...] exerted predominantly inhibitory effects.*" To avoid confusion, the problematic statement was modified to "*Chronos activation [...] exerted predominantly inhibitory effects in PNs*". To shed further light on this, we followed the reviewer's suggestion and analyzed the activity of FS and PN units. Here, it should be noted that we used MUA for our analyses because PNs have very low firing rates, thus requiring a high number of cells or much longer recordings than available in our behavioral experiments. While the yield of well-isolated single units is low with the type of fixed electrodes we used, we found 21 single units. To further address the reviewer's concern, **Review figure 1** shows examples of single unit

responses to brief light pulses. As shown in these representative examples, in response to blue light stimuli, FSs show a strong brief excitation followed by inhibition, whereas PNs show inhibition.

Example firing rate response to 2 ms light pulses

Examples of FS cell phase preference modulation

Review figure 1. (Left) Response of single units recorded in the BLA (Top, FS cell; Bottom, PN) to a single 2-ms blue light stimulus (Average of 65,345 pulses for FS cell and 68,012 pulses for PN cell, **Right**) Gamma entrainment of FS cell (deviation from baseline mean) in the three groups (light stimuli applied at trough - left, peak - middle, or at random times - right). The top row shows entrainment of the cells when the light is off, and the bottom rows show the same cells during light modulation in different phases.

REVIEWER COMMENT 6: Figure 6: If the average triggering frequency is ~20 Hz, and gamma bursts are being filtered around 55 Hz, it means that stimulation is active about 1/3 of the time. How does this compare to the relative amount of time the amygdala spends in a putative gamma state under baseline conditions? Is this value consistent across subjects?

AUTHORS' RESPONSE: To address the reviewer's question, we performed an additional analysis on the baseline BLA recordings after training in the HB task. Specifically, we computed the Z-scored amplitude of the LFP (filtered between 40 and 70 Hz) and then determined the amount of time that the signal was in the different Z-scored bins. The results of this analysis are shown in **Review Figure 2** where the blue bars show the average of all subjects and the grey lines show individual subjects. On average, the amount of time spent in gamma values from 0 to 3 Z-Scores is $39.69 \pm 7.6\%$, which is roughly 1/3 of the time. The grey lines confirm that values are fairly consistent across subjects.

Average time in mid-gamma amplitudes after HB training

Review figure 2. Distribution of gamma amplitude in time. BLA LFP after training on HB task was z-scored and percent time spent in the different bins was computed for each subject (gray lines) and averaged (blue bars).

REVIEWER COMMENT 7: Figure 7: The bidirectional change in gamma power between 40 and 70 Hz seems to be primarily mediated by changes in gamma burst amplitude. As stated previously, reporting similar metrics in Figure 1 would be beneficial.

AUTHORS' RESPONSE: We agree. As detailed in our response to the reviewer's first comment, this information is now included in the revised manuscript.

REVIEWER #2

REVIEWER'S GENERAL COMMENTS: In this manuscript, Kanta et al., develop a method to accurately detect and modulate gamma oscillations, in freely moving rats. Using this method, they establish the causal relationship between post-learning baso-lateral amygdala (BLA) gamma oscillations and consecutive performance during memory retention test of a contextual appetitive task (referred by the authors as HB). The present data are good quality offering a new method for the community and knowledge about BLA physiology. However, I think that the authors could exploit more the data (these data are rare in BLA) and clarify some statements. Please find my comments organized as follow: general and figure-by-figure ones.

REVIEWER COMMENT 1/ The authors develop a method to accurately manipulate gamma

oscillations. However, the authors use gamma power as unique measure to characterize gamma oscillations during behavior (in figure 1). The closed-loop method is doing more than affecting gamma power. The authors could establish the correlation between of mind-gamma events measure (number, strengths and durations) or other measures and consecutive performance during memory retention test, especially for HB task.

AUTHORS' RESPONSE: As requested, we have analyzed the relation between behavioral performance and various properties of gamma bursts (duration, amplitude, incidence) in the IA and HB tasks. These results are shown in a new supplementary figure (**Supplementary Fig. 2** in the updated manuscript). Consistent with the much larger increase in gamma power in the IA than in the HB task, we found a significant correlation between behavioral performance and gamma burst incidence (but not duration or amplitude) in the IA task but not in the HB task. As to the reviewer's second request, **Supplementary Fig. 1f** shows the gamma frequency with the largest power increase in the IA (left) and HB (right) tasks. No obvious difference can be detected between the good and poor learners or between the two tasks. Similarly, no difference was found between the half-width of the frequency band showing the largest increase between the poor and good learners or between the two tasks (**Supplementary Fig. 1g**).

REVIEWER COMMENT 2/ The authors infer from the presence of gamma oscillations in BLA after aversive and appetitive trainings, that consolidation of both task share similar brain mechanism, but this is an over statement. Regretfully, they didn't manipulate gamma oscillations after contextual fear task training (referred by the authors as IA), despite the fact that BLA gamma oscillations after IA training are strong and correlate with consecutive performance during memory retention test. Is the presence of IA data in the paper necessary? Indeed, those data do not support the main conclusion of the paper regarding the role of BLA gamma during consolidation (since it has not been tested) and is not use to test the validity of the closed-loop method (as suggested by the manuscript title). An example of such confusing situation where the authors mix results obtain with IA or HB task, can be read in the lines 248 to 254 of the discussion section. The authors are not able to establish correlation between pre-to-post training changes in mid-gamma and consecutive performance for the HB task, but they are for the IA task. So how they could "experimentally tested the functional importance of these changes in gamma".

AUTHORS' RESPONSE: We respectfully disagree with the reviewer. Both tasks show a boost in gamma power at the same time point during consolidation, and in a similar frequency band. While the increase in gamma power is less pronounced in the appetitive task, this is expected because foraging for food in a familiar environment is a less arousing experience than receiving an unsignaled footshock. True, we found no correlation between post-training gamma power and retention in the appetitive task without light stimulation. However, this is likely because the change in gamma power was lower in this task than in the IA task where a wider range of individual variations was observed. Expanding the range of variations in gamma power through peak or trough optogenetic light stimuli uncovered this correlation in the appetitive task (**Figure 8c**).

REVIEWER COMMENT 3/ This is a general comment for all figures containing electrophysiology. It would great to see raw LFP (as it is generally done in papers from Pare's lab). The authors use raw LFP to show phase detection only. Also they use the same example for figures 3 and 6.

AUTHORS' RESPONSE: While it is true that not all our original figures featured LFPs, raw or filtered LFPs were present in **figures 3a, 3f and 6a**. Most of the other figures without LFPs did

not show actual traces because the figures did not lend themselves to such illustrations. For instance, **Figure 5** did not include LFPs because it illustrates in vitro experiments where there is next to no spontaneous activity. Nevertheless, as requested, we have added actual LFPs wherever we could fit them. Specifically, revised **figure 4** now illustrates five examples of gamma bursts with one, two, three, four, or five triggers. In all five cases, we show raw (black) and superimposed bandpass filtered (red) traces. Furthermore, the new **Supplementary Fig. 3** shows raw LFPs from our 8x8 silicon probe recordings. Also, the repeat LFP in **figure 6** was replaced with an idealized example of a gamma burst, since the purpose of this panel is to illustrate a method.

REVIEWER COMMENT 4/ Comment related to Figure1: Regarding IA, the authors related the presence of post-training BLA gamma oscillations to memory consolidation by citing previous studies (line 49, 4 studies, not really conclusive from my point of view), ignoring studies linking BLA gamma oscillations to fear expression (from Joshua Gordon, Cyril Herry and other labs). As mention earlier, without manipulation of gamma with the closed-loop method, the authors cannot determine the role of post-learning BLA gamma oscillations in IA task consolidation. One plausible explanation is that post-learning BLA gamma oscillations correlated with fear behavior. This could explain also the difference between poor and good learners. The authors should tackle the following question: What are the rats doing during high BLA gamma (that lasted around 20 min) after training? Instead of plotting the average % of time spend in the 3 different sleep-wake states in Supplementary figure 1c, the authors could show how the behavior (the 3 sleep-wake states for example) distribute along the post-training session, for IA and HB tasks.

AUTHORS' RESPONSE: We followed all of the reviewer's suggestions. First, we added references linking BLA gamma to fear expression (lines 50 to 51). Second, we compared the behavior of good and poor learners following training on the IA task. We used two methods to analyze their behavior. We compared the electromyographic (EMG) activity and motion of the rats (assessed by comparing the distribution of light pixels in successive frames of the video files). For both analyses, we integrated the data acquired in 1-min windows. As shown in **Supplementary Fig. 1d,e**, both analyses indicate that good and poor learners are not freezing immediately after IA training. They are similarly mobile and gradually quiet down with time. Thus, it seems unlikely that the difference in gamma power seen between poor and good learners is related to fear behavior as expressed by freezing.

REVIEWER COMMENT 5/ Comment related to Figure 2: In this figure the authors claim that BLA gamma is locally generated. However the only evidence comes from the fact that gamma power is stronger in BLA than CEA and STR. This is not enough. Indeed, CEA and STR are not the only structures surrounding BLA from which gamma could come from. The authors could exploit more the 8x8 silicon probes used. I couldn't find any details about those probes and recordings, like: length of shanks, space between shanks and histological locations. The authors show phase-coherence along the dorso-ventral axis (is that along all 8 recording sites?) of BLA but not in the medio-lateral axis. What is the phase-coherence between BLA, CEA and STR? More importantly, the authors have to show that locally recorded BLA gamma entrain local units and not just cite previous work.

AUTHORS' RESPONSE: We have added the requested details about the silicon probe configuration, and these are now provided in the methods section (lines 336-338).

Regarding other sites besides CeA and STR, the edges of our silicon probes did partially extend into the adjacent cortices. Unfortunately, because these only included a single row of

electrodes (either the lateral-most shank or the bottom-most recording sites across shanks), their CSDs are subject to distortion from edge effects and the known dipole structure of electric fields produced by cortical pyramidal cells. This issue requires us to exclude these sites from those analyses. These caveats do not apply to LFPs, and panels b, c, d, and e of Figure 2 include cortical sites, and the phase coherence plots have been updated to include eight sites aligned along both the dorsal-ventral and medial-lateral axes. As one would expect, cortical regions exhibit gamma oscillations (**Figure 2b,c**), but notably these tended to have reduced phase-coherence with the BLA (**Figure 2d,e**), and in those panels it is evident that the same was true for sites in the CeA/STR.

To address Reviewer 2's suggestion that BLA unit activity should be entrained to gamma, we measured multi-unit activity at BLA sites and calculated their firing rate modulation as a function of gamma phase at an adjacent site also in the BLA (**Figure 2a**). Unit activity was modulated by gamma phase, with spiking preferentially occurring during the trough of the gamma oscillation, which is typical when unit activity is modulated by a gamma rhythm arising from local synaptic activity.

While we cannot rule out the possibility that volume conduction from surrounding sites contributes to some extent to the gamma oscillations seen in the BLA, the fact that BLA neurons are strongly and consistently entrained to a specific phase of the gamma cycle indicates that local synaptic currents do occur rhythmically at the gamma frequency in the BLA. The logically inescapable consequence of these findings is that the LFP is a reliable indicator of the cyclical shifts in neuronal excitability occurring in the BLA during gamma oscillations. That's all that is needed for our closed-loop method of gamma modulation to be valid.

REVIEWER COMMENT 6/ Comment related to Figure 5: I think that the goal of this figure should be to evaluate the effect of the stimulations used in vivo on BLA neurons. However the authors describe how neurons response to single light pulses, in vitro and in vivo. There is only one example of PN response for light pulse-train around 20hz, not gamma range! They use Chronos to maximize the temporal precision of the method, they should show it. Because of the model (rats) the authors use a non-specific optogenetic strategy, it's fine, but from this figure we don't now what is the activity of neurons during gamma modulations, in vitro and in vivo.

AUTHORS' RESPONSE: As requested, the revised manuscript now shows the responses of FSs and PNs to trains of light stimuli in the gamma range (50 Hz) in vitro (New **Supplementary Fig. 6**). Furthermore, **Review figure 1** shows example single unit responses to brief light pulses (left) and contrasts their entrainment by gamma in the different light groups (right).

REVIEWER COMMENT 7/ Comment related to Figure 8: This is the most important figure, validating the method. It's really interesting results, regretfully, we don't know if it's working for IA task, even with inter-subject comparisons. 8c is confusing to me, why pulling together peak and trough groups for the correlation calculation? It makes more sense to separate the two groups. Peak group don't show correlation between gamma power and performance, as in figure 1, but the trough group show high correlation.

AUTHORS' RESPONSE: As now explained in the revised figure legend, the Trough and Peak groups have been combined to show that an expanded range of gamma levels (caused by the modulation) unveiled a linear correlation with performance. When the various conditions are

considered separately, the range of gamma levels is too narrow to correlate with behavior, as it was in our baseline recordings (**Figure 1m**).

REVIEWERS' COMMENTS:

Reviewer #1 (Remarks to the Author):

The manuscript has improved greatly after addressing the reviewers' comments. In particular, the revised version of Figure 2 is much clearer.

It is helpful to see additional analysis of power spectra characteristics in Supplementary Figure 1. However, the authors should still add a legend to panel A to indicate which type of subjects each line came from.

The authors should consider revising Figure 5h to include more details about the breakdown of response types among spiking PNs found in Supplementary Figure 6, which I found very helpful. It would also be justified to move Review Figure 1 to the manuscript. Even though there are not many single units included, it's still helpful to see the light-driven FS activation and PN inhibition is not just a consequence of in vitro slice conditions.

The new title is better, but is still quite vague, and doesn't really capture the essence of the findings. The authors might consider adding some more relevant details, such as "Closed-loop control of amygdalar gamma oscillations demonstrates their role in the consolidation of spatial memories."

Reviewer #2 (Remarks to the Author):

The authors addressed all my concerns. The study is good quality offering a new method for the community and knowledge about amygdala physiology.

REVIEWERS' COMMENTS:

Reviewer #1 (Remarks to the Author):

The manuscript has improved greatly after addressing the reviewers' comments. In particular, the revised version of Figure 2 is much clearer.

AUTHOR'S RESPONSE: We thank the reviewer for their suggestions that helped us improve the manuscript.

It is helpful to see additional analysis of power spectra characteristics in Supplementary Figure 1. However, the authors should still add a legend to panel A to indicate which type of subjects each line came from.

AUTHOR'S RESPONSE: We followed the reviewer's suggestion and now indicate the types of subjects in the figure legend.

The authors should consider revising Figure 5h to include more details about the breakdown of response types among spiking PNs found in Supplementary Figure 6, which I found very helpful. It would also be justified to move Review Figure 1 to the manuscript. Even though there are not many single units included, it's still helpful to see the light-driven FS activation and PN inhibition is not just a consequence of in vitro slice conditions.

AUTHOR'S RESPONSE: We revised Figure 5 to now include the pie charts from Supplementary Figure 6 (revised **Figure 5i**). We also added the two examples from Review Figure 1 into Figure 5 (revised **Figure 5j,k**).

The new title is better, but is still quite vague, and doesn't really capture the essence of the findings. The authors might consider adding some more relevant details, such as "Closed-loop control of amygdalar gamma oscillations demonstrates their role in the consolidation of spatial memories."

AUTHOR'S RESPONSE: Following the reviewer's suggestion, we changed our title to:

"Closed-loop control of gamma oscillations in the amygdala demonstrates their role in spatial memory consolidation"

Reviewer #2 (Remarks to the Author):

The authors addressed all my concerns. The study is good quality offering a new method for the community and knowledge about amygdala physiology.

AUTHOR'S RESPONSE: We thank the reviewer for their recommendations that helped us improve the manuscript.